# SPECULATIVE SPECULATIVE DECODING

**Tanishq Kumar**[1]* **Tri Dao**[2,3] **Avner May**[3]

[1]Stanford University    [2]Princeton University    [3]Together AI

## ABSTRACT

Autoregressive decoding is bottlenecked by its *sequential* nature. Speculative decoding has become a standard way to accelerate inference by using a fast draft model to predict upcoming tokens from a slower target model, and then verifying them *in parallel* with a single target model forward pass. However, speculative decoding itself relies on a *sequential* dependence between speculation and verification. We introduce *speculative speculative decoding* (SSD) to parallelize these operations. While a verification is ongoing, the draft model *predicts* likely verification outcomes and prepares speculations pre-emptively for them. If the actual verification outcome is in the predicted set, a speculation can be returned immediately, eliminating drafting overhead entirely. We identify three key challenges presented by speculative speculative decoding, and suggest principled methods to solve each. The result is SAGUARO, an optimized SSD algorithm. Our implementation is on average 30% faster than the strongest speculative decoding baselines and up to 5x faster than autoregressive decoding with open source inference engines.

## 1 INTRODUCTION

Modern AI demands fast inference. Yet standard language model decoding generates single tokens sequentially, failing to leverage the massive parallel computation available on modern hardware. Speculative decoding (Leviathan et al., 2023; Chen et al., 2023) (SD) is a technique introduced to alleviate this problem. Instead of slow, autoregressive sampling from a desired "target model," it uses a fast "draft model" to predict the next few tokens that would be generated by the target model and then "verifies" these tokens in one *parallel* forward pass of the target model. This verification is done according to an algorithm that guarantees the resulting tokens are drawn from the distribution of the target model. In each verification, the target model decides both how many speculated tokens to accept, *and* samples an additional *bonus token* that follows all of the accepted tokens. Though speculative decoding is effective, it is *itself* limited by a *sequential* dependence: verification must complete before the next speculation can begin. We ask:

*Can we eliminate the sequential dependence between drafting and verification?*

We introduce *speculative speculative decoding* (SSD), a unifying framework for methods that aim to parallelize drafting and verification. While in SD the draft model *waits* for the verification to complete before beginning to speculate the next round, in SSD the draft model *predicts* what verification outcomes are most likely, and prepares (i.e., pre-speculates) for all of them *in parallel* to the verification taking place. If any of these prepared outcomes occurs, the draft model can *immediately* send the pre-speculated tokens to the verifier, thereby avoiding all draft overhead. Like ordinary speculative decoding, SSD is lossless. Unlike ordinary speculative decoding, in SSD the draft model is located on distinct hardware from the target.

There are three main challenges in optimizing SSD algorithms. First, the draft model must correctly anticipate the verification outcome, which includes not just how many speculated tokens were accepted, but *which* bonus token was sampled. Second, we identify a subtle trade-off between the acceptance rate of the speculator and how well it is able to predict verification outcomes, which must be navigated carefully to maximize speedups. Third, any SSD algorithm must have a fallback strategy to handle failed attempts at predicting verification outcomes correctly. At large batch sizes and

---

*Correspondence to `tanishq@stanford.edu`. Code at `github.com/tanishqkumar/ssd`.

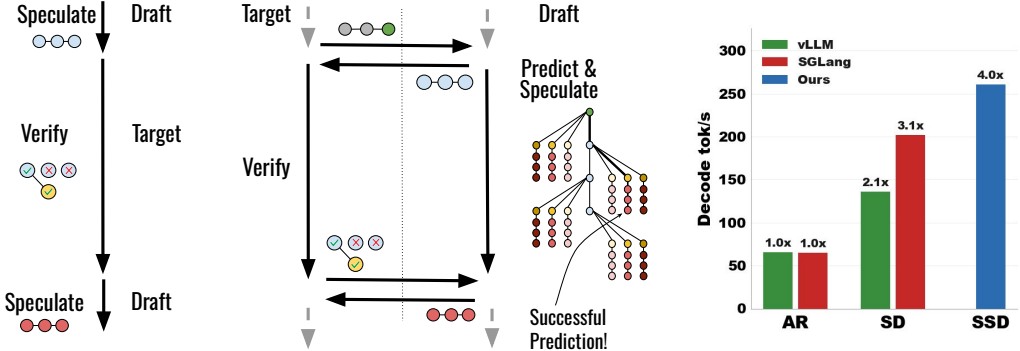

Figure 1: (Left) Ordinary speculative decoding (SD) requires the verifier to wait idly for the draft to speculate. (Center) In our algorithm, speculation runs on a separate device (1×H100) in parallel with verification; the draft precomputes speculations for many possible verification outcomes and returns the speculated tokens immediately if one occurs. (Right) *End-to-end performance* of SSD, SD, and autoregressive (AR) decoding averaged over four datasets spanning math, code and chat, for Llama-3.1-70B on TP=4 H100s, batch size 1, greedy decoding, Llama-3.2-1B draft model.

temperatures, these failures happen increasingly often and naïvely speculating just-in-time negates the benefits of asynchrony.

We present SAGUARO, an optimized SSD algorithm that incorporates targeted optimizations for each challenge:

- In Section 4.1, we frame the problem of predicting verification outcomes in terms of constrained optimization, and introduce a technique that uses the most likely draft logits to predict the bonus token, doing so with up to 90% accuracy.
- In Section 4.2, we identify a tension between accurately predicting verification outcomes and generating high-quality speculations, and develop a sampling algorithm that allows balancing the two. Appendix A.3.1 gives a construction where this new sampling scheme necessarily yields end-to-end speedups.
- In Section 4.3, we examine various strategies to handle failed predictions, demonstrating that the optimal fallback strategy varies with batch size. Adopting this, SAGUARO outperforms SD 20% even at larger batch sizes, despite doing more compute per batch element by decoding many possible outcomes simultaneously.

**Altogether, SAGUARO is on average 30% faster than the strongest speculative decoding baselines and up to 5x faster than autoregressive generation (Figure 7), while strictly improving the throughput-latency Pareto frontier across a range of batch sizes.**

## 2 BACKGROUND

We briefly review speculative decoding (Section 2.1) and related work (Section 2.2) to provide important background and context for our work.

### 2.1 SPECULATIVE DECODING

Speculative decoding accelerates sampling from a target distribution $p_{\text{target}}$ by using a cheaper draft distribution $p_{\text{draft}}$ to generate candidate continuations, and then using the target distribution to accept selectively. To verify, the target first computes probabilities for all the speculated tokens in parallel in *a single forward pass*. Drafted tokens are accepted sequentially, each with probability $\min\{1, p_{\text{target}}(x)/p_{\text{draft}}(x)\}$. Upon the first rejection, remaining draft tokens are discarded, and the last-token target logits are used to sample a "bonus token." This is done by sampling the bonus token from a modified distribution that guarantees the resulting sequence is distributed as $p_{\text{target}}$. This modified distribution is called the ***residual distribution***, and takes the form

$r(\cdot) \propto \max\left(p_{\text{target}}(\cdot) - p_{\text{draft}}(\cdot), 0\right)$. In the case where all tokens are accepted, the residual distribution is simply the target distribution.

The expected number of generated tokens per round, and thus the overall efficiency of speculative decoding, is governed by the ***acceptance rate***: the probability of accepting a given token in the speculation, conditioned on accepting all prior tokens. In (Leviathan et al., 2023), it is shown that the acceptance rate can be written in terms of how well the draft distribution approximates the target distribution in the following way.

**Theorem 1.** *(Leviathan et al., 2023)*

$$\alpha = \sum_x \min\{p_{\text{target}}(x), p_{\text{draft}}(x)\} = 1 - \tfrac{1}{2}\|p_{\text{target}} - p_{\text{draft}}\|_1$$

**Definition 2.** *We define a **speculation** at round $T$, denoted $s^T := (s_1^T, \ldots, s_K^T)$, as the sequence of $K$ tokens proposed autoregressively by the draft model. The length $K$ of the speculated sequence is called the **speculative lookahead**.*

**Definition 3.** *We define a **verification outcome** at round $T$, denoted $v^T := (k, t^*) \in \mathcal{V}^T$, where $(s_1^{T-1}, \ldots, s_k^{T-1})$ are the accepted draft tokens from round $T-1$, and $t^*$ is the **bonus token** sampled from the residual distribution (or target distribution if all tokens are accepted).*

## 2.2 RELATED WORK

**Parallel Speculative Decoding.** AMUSD (McDanel, 2025) and PEARL (Liu et al., 2025) propose speculating the next round during ongoing verification, but only prepare for the verification outcome in which all tokens are accepted, which is one special case of many possible outcomes. DSI (Timor et al., 2025) also overlaps drafting and verification, verifying multiple sequences in parallel with distinct logical copies of the verifier model. Mirror-SD (Bhendawade et al., 2025) includings branch-complete speculative rollouts conditioned on early-exit signals from the target model and heterogeneous execution across devices. SwiftSpec (Zhang et al., 2025) and SpecBranch (Shen et al., 2025) prepare a larger cache consisting of a token tree branching off of the speculation being verified (thus enabling larger speedups), but both use fallback strategies that do not work at large batch sizes. SpecBranch (Shen et al., 2025) constitutes (approximately) a special case of the SSD framework where only a single branching point is allowed and where the fallback speculator is equal to the regular speculator. SwiftSpec (Zhang et al., 2025) considers the special case of greedy sampling, and adopts a fallback strategy of just-in-time speculation that struggles at higher temperatures and batch sizes when cache-misses become inevitable.

**Tree-Based Speculative Decoding.** Numerous speculative decoding methods have been proposed that increase the expected number of accepted tokens by allowing the draft model to speculate a *tree* of tokens instead of a sequence, thereby giving the verifier several token options at each position (Miao et al., 2024; Li et al., 2024b; Chen et al., 2024; Svirschevski et al., 2024). Our method differs in important ways: First, existing tree-based methods are still sequential: speculate, then verify. Second, tree-based methods introduce a large amount of *verifier compute*—which is quite expensive due to the size of the target model—because the entire tree must be verified in the target model forward pass. Our method, on the other hand, scales up the *speculation compute* by pre-speculating for many verification outcomes in parallel, but does not introduce any additional verifier compute. Lastly, it is important to note that our method can be combined with these tree-based approaches for further gains (see Appendix E for discussion).

**Improved Draft Architectures.** There have been many advancements in draft model architectures that can improve the acceptance rates and/or speeds of the draft model and which can be used in SSD for even better performance. For example, EAGLE (Li et al., 2024a;b; 2025b) allows the draft model to take as input the powerful representations from the target model for the current prefix, while GliDe (Du et al., 2024) and LongSpec (Yang et al., 2025) allow the draft model to perform cross-attention on the KV cache of the target model. Alternate model architectures, like diffusion LLMs (Nie et al., 2025; Sahoo et al., 2024; Li et al., 2025a; Christopher et al., 2025; Samragh et al., 2025; Chen et al., 2026) or SSMs (Gu et al., 2022; Gu & Dao, 2023; Wang et al., 2024), can also be used to increase the speed with which the draft model can produce the speculated token sequence. We include more technical details on how SSD can fruitfully be combined with these improved draft model architectures (e.g., EAGLE-3; Li et al. (2025b)) in Appendix E.

## 3 THE SPECULATIVE SPECULATIVE DECODING FRAMEWORK

We introduce *speculative speculative decoding*, a framework to reason about asynchronous variants of speculative decoding (Section 3.1), and present results comparing the expected speed of SSD to baselines (Section 3.2). **Definitions introduced here will be important in Section 4.**

### 3.1 ALGORITHM

We present the SSD framework in Algorithm 1. The speculator and verifier processes run in parallel on separate hardware. While the verifier is verifying the drafted tokens from round $T$, the speculator begins speculating round $T + 1$. It does so by predicting likely verification outcomes, and then preparing speculations for each of these outcomes in parallel, and storing these in a "speculation cache" (defined formally below). Then, when it receives the actual verification outcome, it checks whether this was one of the outcomes in the cache that it had prepared for. If so, it immediately returns it. Otherwise it defers to a fallback speculation strategy.

For intuition on the SSD algorithm and why it is lossless, consider the similarities with speculative execution (González & González (1997)) for CPU optimization. Like speculative execution—which uses idle compute to pre-execute conditional paths in case they are later needed—SSD pre-speculates over many possible verification outcomes so that, when one occurs, the corresponding token sequence is immediately available. That sequence is then verified exactly as in SD. If the realized outcome was not pre-speculated, SSD falls back to a synchronous speculator, which is lossless since it reduces to ordinary SD. The following definitions will be important in understanding the SSD framework.

**Definition 4.** *We define the **speculation cache** $\mathcal{S}^T$ for round $T$ as a dictionary mapping from a set of verification outcomes $\mathcal{V}^T$ to precomputed speculations for those outcomes. We denote a speculated token sequence $s^T$ corresponding to an outcome $v^T \in \mathcal{V}^T$ by $\mathcal{S}^T(v^T)$ as a cache lookup operation.*

**Definition 5.** *A **cache hit** is when the outcome $v^T$ of verifying $s^{T-1}$ is contained in the speculation cache $\mathcal{S}^T$. A **cache miss** is when $v^T \notin \mathcal{S}^T$.*

**Definition 6.** *Let $\mathbf{p}_{\mathrm{hit,p}}$ denote the probability of a cache hit on an iteration conditional on the previous iteration having been speculated by the primary speculator. Analogously, $\mathbf{p}_{\mathrm{hit,b}}$ represents the probability of a cache hit on an iteration conditional on speculating with the backup model. Finally, let $\mathbf{p}_{\mathrm{hit}}$ denote the unconditional probability of a cache hit in an iteration.*

### 3.2 THEORETICAL RESULTS

We analyze the performance of SAGUARO relative to regular autoregressive decoding (Theorem 7) and ordinary speculative decoding (Corollary 9). Proofs deferred to Appendix A.

**Theorem 7.** *Let the primary and backup speculators take time $T_p$ and $T_b$ relative to the verifier. Let $E_{\mathrm{hit}}$ and $E_{\mathrm{miss}}$ denote the expected number of generated tokens from the primary and backup speculators, respectively. The expected speedup of Algorithm 1 relative to autoregressive decoding is then:*

$$speedup_{\mathrm{SSD}} \quad = \quad \frac{p_{\mathrm{hit}} \cdot E_{\mathrm{hit}} + (1 - p_{\mathrm{hit}}) \cdot E_{\mathrm{miss}}}{p_{\mathrm{hit}} \cdot \max(1, T_p) + (1 - p_{\mathrm{hit}}) \cdot (1 + T_b)}.$$

The numerator corresponds to the expected number of tokens generated in each iteration of the algorithm, and the denominator corresponds to the expected latency of each iteration relative to autoregressive decoding. This implies two key corollaries.

**Corollary 8.** *(**Strictly Faster Than SD**) Suppose we run SD with a given speculator $\mathcal{M}$. Running SSD with primary and backup both set to $\mathcal{M}$ does no worse than SD (strictly better if $p_{\mathrm{hit}} > 0$, $T_{\mathrm{SD}} > 0$).*

**Corollary 9.** *(**Speedup Sandwich**) Suppose we choose a primary speculator for which drafting completes before verification ($T_p < 1$), and a fast backup speculator ($T_b = 0$). Then if $T_{\mathrm{SD}}, E_{\mathrm{SD}}$ represent the latency and expected number of generated tokens from a draft model in SD, then the SSD speedup over SD can be bounded by:*

$$\left(1 + T_{\mathrm{SD}}\right) \cdot \frac{E_{\mathrm{hit}}}{E_{\mathrm{SD}}} \cdot p_{\mathrm{hit}} \leq \frac{speedup_{\mathrm{SSD}}}{speedup_{\mathrm{SD}}} \leq \left(1 + T_{\mathrm{SD}}\right) \cdot \frac{E_{\mathrm{hit}}}{E_{\mathrm{SD}}}.$$

---

**Algorithm 1:** The Speculative Speculative Decoding (SSD) Framework.

---

**Function** main (*prompt, target, primary_draft, backup_draft*) :
  asynchronously launch speculator (*prompt, primary_draft, backup_draft*)
  $generated\_tokens \leftarrow$ verifier (*prompt, target*)
  **return** $generated\_tokens$

**Function** verifier (*prompt, target*) :
  $target$.prefill (*prompt*)
  **WAIT TO RECEIVE** $spec\_tokens$ from speculator
  $generated\_tokens \leftarrow []$
  **while** *True* **do**
      $verify\_outcome \leftarrow target$.verify (*spec_tokens*)
      $generated\_tokens$.append (*verify_outcome.tokens*)
      **SEND** $verify\_outcome$ to speculator
      **if** *end_token* $\in verify\_outcome$ **then**
          **return** $generated\_tokens$
      **end**
      **WAIT TO RECEIVE** $spec\_tokens$ from speculator
  **end**

**Function** speculator (*prompt, primary_draft, backup_draft*) :
  $primary\_draft$.prefill (*prompt*)
  $spec\_tokens \leftarrow primary\_draft$.speculate (*prompt*)
  **while** *True* **do**
      **SEND** $spec\_tokens$ to verifier for verification
      $outcomes \leftarrow$ predict_verify_outcomes (*spec_tokens, primary_draft*) // Sec. 4.1
      $cache \leftarrow$ speculate_for_outcomes (*outcomes, primary_draft*) // Sec. 4.2
      **WAIT TO RECEIVE** $verify\_outcome$ from verifier
      **if** *end_token* $\in verify\_outcome$ **then**
          **return**
      **end**
      **if** $verify\_outcome \in cache$ **then**
          $spec\_tokens \leftarrow cache[verify\_outcome]$
      **else**
          $spec\_tokens \leftarrow$ fallback_speculate (
              verify_outcome, primary_spec, backup_spec
          ) // Sec. 4.3
      **end**
  **end**

---

This equation reveals that the maximum speedup attainable by SSD is proportional to the latency reduction $(1 + T_{\text{SD}})$ from hiding draft latency, and the increase in expected number of generated tokens $(E_{\text{hit}}/E_{\text{SD}})$ from increased drafting time. However, a low cache hit rate $p_{\text{hit}}$ reduces the effectiveness of this algorithm, as shown by the lower bound.

## 4 SAGUARO: AN OPTIMIZED SSD ALGORITHM

In this section, we present SAGUARO, our optimized instantiation of the SSD framework. We present the three core optimizations in Sections 4.1 (SAGUARO cache construction), 4.2 (SAGUARO sampling), and 4.3 (SAGUARO fallback), respectively, then evaluate SAGUARO end-to-end in Section 5.

**Setup.** Unless otherwise specified, experiments are run with batch size 1 and greedy decoding, with the target model on 4×H100. In SD, the draft is collocated on the same hardware. In SSD, it sits on a separate device (1×H100), since it speculates asynchronously during verification. We study two model families, Llama-3 (main text) and Qwen-3 (Appendix F) across four datasets: Alpaca (Dubois et al., 2023), GSM8k (Cobbe et al., 2021), UltraFeedback (Cui et al., 2024), HumanEval (Chen et al., 2021). Further details in Appendix B.

### 4.1 PREDICTING VERIFICATION OUTCOMES: BUILDING THE SAGUARO CACHE

Given a speculation $s^T$ that is in the process of being verified, SAGUARO must build a cache $\mathcal{S}^T$ for the most likely verification outcomes. The difficulty is that the space of possible verification outcomes is vast, of size approximately $(K+1)V$ where $V$ is the model vocabulary size. Since there are $O(KV)$ potential verification outcomes, it is impractical to preemptively speculate for *all* of them in parallel. This motivates posing verification prediction as constrained optimization: given a budget of $B$ verification outcomes, how should one select them to maximize the chances of a cache hit? Note that the budget $B$ typically corresponding to the maximum number of outcomes the draft can finish speculating for before verification completes.

#### 4.1.1 ALGORITHM

**Definition 10.** *We define the **fan-out** $F_k^p := \left| \{ v^T := (k', t^*) \in \mathcal{S}^T \ | \ k' = k \} \right|$ at position $k$ to be the number of verification outcomes with $k$ accepted tokens that are included in the speculation cache, given the previous iteration was speculated by the primary speculator. $F_k^b$ is defined analogously for the backup speculator.*

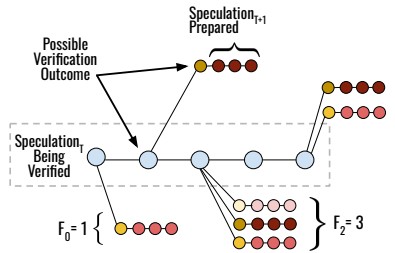

Figure 2: Schematic of speculation cache strategy. We allocate *fan-out $F_k$* (bonus token guesses) over sequence length $K + 1$ according to Theorem 12.

**Saguaro Verification Outcome Prediction Algorithm** Given a fan-out strategy $\{F_k^p, F_k^b\}$ (ours based on Theorem 12), SAGUARO takes the top-$F_k$ tokens at each lookahead position $k$ in the draft logits, and adds those to the speculation cache. These are the top-$F_k$ logits *excluding* the sampled token sent for verification at that position, which is *guaranteed* not to be the bonus token.

#### 4.1.2 THEORETICAL RESULTS

We show how to optimize the choice of fan-out values $F_k^p$ and $F_k^b$ to maximize the speedup under a constraint on the size of the speculation cache.

The probability of a verification outcome $(k, \cdot)$ in a given iteration depends on whether the previous speculation was generated by the primary or backup speculator. Thus, if the default fan-out strategy when building the cache is to fan out uniformly at each sequence position, the cache hit rate on a given iteration depends on whether the previous iteration was speculated by the primary or backup speculator. These are the definitions of $p_{\text{hit,p}}(F)$ and $p_{\text{hit,b}}(F)$, respectively. We see in Figure 3 that these functions (in fact their complement, the rejection rate) empirically follow a power-law.

**Definition 11.** [1] *We say that a speculator has a **r power-law cache hit rate** if the chance of a cache miss with fan-out $F$ is equal to a power-law of $F$ with exponent $r$, for a sequence drafted by this speculator. More specifically, this means $1 - p_{\text{hit},*}(F) = 1/F^r \ \ \forall F \in \mathbb{N}$, for some $r > 0$.*

We now use this definition to reason about how to optimally select the fan-out values $F_k^p$ and $F_k^b$ under a computational constraint on the size of the cache. We consider the general case, when this assumption doesn't hold, in Appendix A.

**Theorem 12.** *(SAGUARO **Cache Shape: Geometric Fan-Out**) Consider a draft model with acceptance rate $a_p$ and a r power-law cache hit rate. Then the optimal choice of $F_k^p$ values for $k \in [0, K]$ under the constraint $\sum_{k=0}^{K} F_k^p \leq B$, follows a capped geometric series:*

$$F_k = F_0 \cdot a_p^{k/(1+r)} \ \ \forall k < K, \text{ and}$$
$$F_K = F_0 \cdot a_p^{K/(1+r)} \cdot (1 - a_p)^{-1/(1+r)},$$

*where $F_0$ can be chosen such that $\sum_{k=0}^{K} F_k^p = B$ (closed form-equation in Appendix A). An equivalent result holds for $F_k^b$.*

---

[1] This definition is closely related to the notion of $b$ power-law acceptance rate in Chen et al. (2024).

This result cleanly reflects the intuition that the lengths of verified strings follow a capped geometric distribution supported on the speculation lookahead. In other words, if it is unlikely that $j \in [0, K]$ tokens will be accepted by the verifier, we should not waste compute guessing and speculating on the bonus token at that position, and should lower $F_j$.

### 4.1.3 EMPIRICAL EVALUATION

We compare the end-to-end performance of using a naive (uniform) fan-out strategy over sequence length against the geometric fan-out strategy advanced by Theorem 12. In Figure 4 we find it improves cache hit rate (right) and end-to-end decoding speed, especially at higher temperatures, where both SD and the uniform fan out begin to flounder.

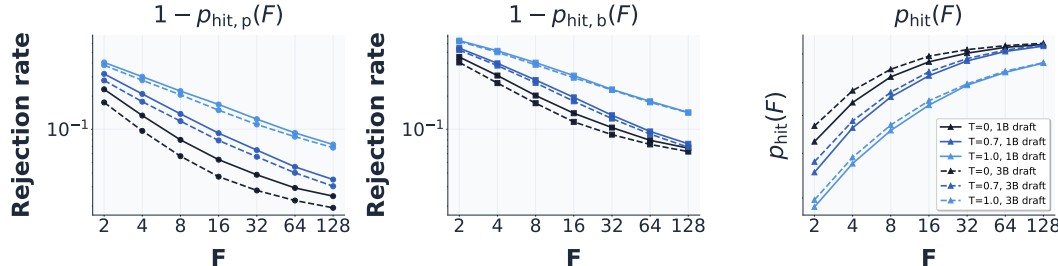

Figure 3: Scaling of cache hit rates with fan out. (Left, middle) The rejection rates $(1 - p_{hit,*}(F))$ conditioned on the prior speculation coming from the primary vs backup speculator, respectively. Rejection rates (cache misses) fall as a power law in the draft-fan out, demonstrating that cache hit rates increase with cache size. (Right) The overall cache hit rate $p_{hit}(F)$.

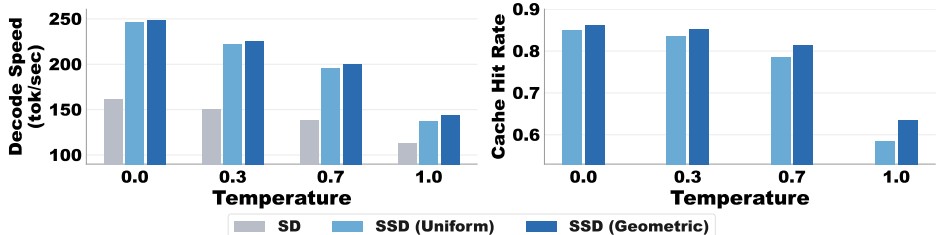

Figure 4: Advantage of geometric fan out strategy increases at higher temperatures, improving both speculation cache hit rate (right) and thus end-to-end speed (left). Results averaged over four datasets. At all temperatures, SSD with either fan out strategy outperforms ordinary speculative decoding.

## 4.2 BALANCING CACHE HIT AND ACCEPTANCE RATE WITH SAGUARO SAMPLING

The majority of the time, the bonus token is sampled from the *residual distribution* (except when all tokens are accepted, in which case it is sampled from the target directly). Recall the residual distribution is given by $r(\cdot) \propto \max(p_{\text{target}}(\cdot) - p_{\text{draft}}(\cdot), 0)$. This distribution can be difficult to predict, especially as sampling temperatures increase. We introduce a novel sampling scheme that makes this residual easier to predict and therefore increases cache hit rates.

This sampling scheme exploits the fact that the residual distribution is a function of the draft distribution (which includes the draft sampling scheme). We make the bonus token easier to predict by explicitly increasing the residual probability mass on the most likely draft tokens; and this is done by *decreasing* the corresponding probability mass when sampling from the draft. In biasing the draft distribution, however, we may decrease the acceptance rate by having moved the draft distribution farther from the target (see 1), inducing a tradeoff between the acceptance rate and the cache hit rate, both of which contribute to end-to-end speed (see Figure 5, left). We show in Appendix A.3.1 that there exist distributions $p_{\text{target}}, p_{\text{draft}}$ for which biasing the draft distribution in this way leads to end-to-end speedups.

### 4.2.1 ALGORITHM

**Definition 13.** *We define a **sampling scheme** as a function $\sigma\colon \mathbb{R}^V \to \Delta^{V-1}$ from model logits $z_{\text{draft}} \in \mathbb{R}^V$ to a probability distribution $p \in \Delta^{V-1}$.*

To understand how to design a sampling scheme that increases the residual probability mass on the most likely draft tokens, we first note that the probability of token $t$ in the residual distribution is proportional to $\max(p_{\text{target}}(t) - p_{\text{draft}}(t), 0)$. Thus, *decreasing* $p_{\text{draft}}(t)$ *increases* the probability of $t$ in the residual. This is exactly what our cache-aware sampling scheme does.

**Definition 14.** *Given draft logits $z \in \mathbb{R}^V$, we define the **SAGUARO** sampling scheme $\sigma_{F,C}(z)$ for fan-out $F$ and downweighting constant $C \in [0, 1]$ as*

$$\sigma_{F,C}(z) \propto \begin{cases} C \cdot \exp\left(z_t\right) & \text{if } t \in top_F(z) \\ \exp\left(z_t\right) & \text{otherwise,} \end{cases}$$

In practice, $C$ is a hyperparameter that is chosen empirically.

### 4.2.2 THEORETICAL RESULTS

**Theorem 15.** *For fan-out $F$ and primary speculator logits $z$, the cache hit rate $p_{\text{hit}}$ of the SAGUARO sampling scheme increases monotonically as $C \to 0$.*

The new sampling hyperparameter $C$ allows a trade-off between cache hit rate and acceptance rate. We plot this tradeoff in Figure 5 (left). Figure 5 (right) presents an illustration of how SAGUARO sampling allows control of the *residual distribution* by manipulating the draft distribution during speculation. The intuition here is that SAGUARO sampling deliberately suppresses the draft probabilities on the $F$ cached tokens. By downweighting $p_{\text{draft}}(t)$ on this set, the residual $\max\big(p_{\text{target}}(t) - p_{\text{draft}}(t),\, 0\big)$ is pushed to *concentrate* on those same tokens, *increasing* the chance that the bonus token lands inside the cache by construction.

### 4.2.3 EMPIRICAL EVALUATION

We show in Figure 5 how there is a trade-off between acceptance rate and cache hit rate which we can navigate by using the SAGUARO sampling scheme with different values of $C$. Lower values of $C$ lead to higher cache hit rates and lower acceptance rates as they bias the draft distribution away from the target to control the residual distribution.

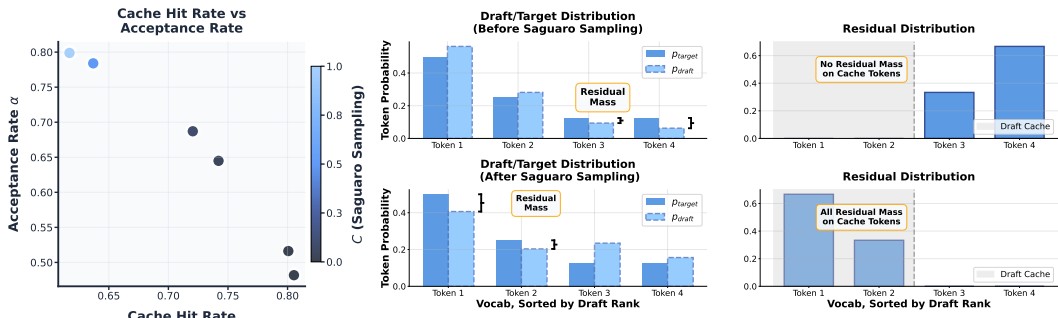

Figure 5: We introduce SAGUARO sampling, a novel sampling scheme designed specifically for SSD. (Left) It interpolates between high cache hit rate and high speculative acceptance rate. (Right) Illustrative schematic for how SAGUARO sampling increases residual probability mass on the top draft tokens, encouraging the sampled bonus token to lie in the speculation cache by construction.

### 4.3 HANDLING CACHE MISSES WITH SAGUARO FALLBACK

We now discuss how we optimize the handling of cache misses in SAGUARO, and propose an optimal strategy for picking the backup speculator based on the batch size.

### 4.3.1 ALGORITHM

We design SAGUARO's cache miss strategy based on the observation that cache misses occur *almost certainly* at large batch sizes, and that when this happens, in SSD the *whole batch* must wait for the backup speculator to complete before being verified. We propose the SAGUARO *fallback strategy*: set the backup speculator to be equal to the primary speculator at low batch size, then switch to a low-latency speculator (Oliaro et al., 2024; Liu et al., 2024; Xu et al., 2025) for larger batch sizes $b > b^*$, where the critical batch size $b^*$ is derived in the following section.

### 4.3.2 THEORETICAL RESULTS

We prove that SAGUARO's backup speculator strategy is optimal, under the conditions that SSD uses a high-quality (slow) speculator (primary), and a lower-quality (fast) speculator (backup). We begin with a corollary to Theorem 7 which accounts for the impact of batch size on the expected speedup attained by SSD. This result assumes that in SSD the whole batch waits for the backup speculator to complete before verifying the batch.

**Corollary 16.** *At batch size $b$, the expected speedup from SSD is equal to:*

$$speedup = \frac{p_{\text{hit}} \cdot E_{\text{hit}} + (1 - p_{\text{hit}}) \cdot E_{\text{miss}}}{p_{\text{hit}}^b \cdot \max(1, T_p) + (1 - p_{\text{hit}}^b) \cdot (1 + T_b)}, \quad \text{which approaches}$$

$$\frac{p_{\text{hit}} \cdot E_{\text{hit}} + (1 - p_{\text{hit}}) \cdot E_{\text{miss}}}{1 + T_b} \quad \text{as } b \to \infty.$$

In our implementation, the backup strategy is to return random tokens,[2] and the primary strategy is to do just-in-time speculation. As the batch size increases, the entire batch stalls on the latency of the backup speculator as cache misses happen more frequently. This forces the choice of a low latency speculator at larger batch sizes, as the following theorem makes precise.

**Theorem 17.** *The optimal fallback strategy for cache misses, given two speculators of varying speeds and quality, is to use the slow and accurate speculator as the backup for batch sizes $b < b^*$, and the fast speculator otherwise. We solve for $b^*$ in Appendix A.*

### 4.3.3 EMPIRICAL EVALUATION

We show in Figure 6 (left) that as the batch size increases, using a fast backup speculator that returns random tokens outperforms using a slow but more accurate neural speculator just-in-time. In Figure 6 (right) we show that we can reduce latency further by scaling draft-time compute. Specifically, we show the projected speedups we would get at batch size 16—computed based on our cache hit rate curves (Figure 4)—if we increase the number of draft GPUs (and thus, the fan-out factor). Dedicating more devices to speculation leads to larger speculation caches and thus higher cache hit rates.

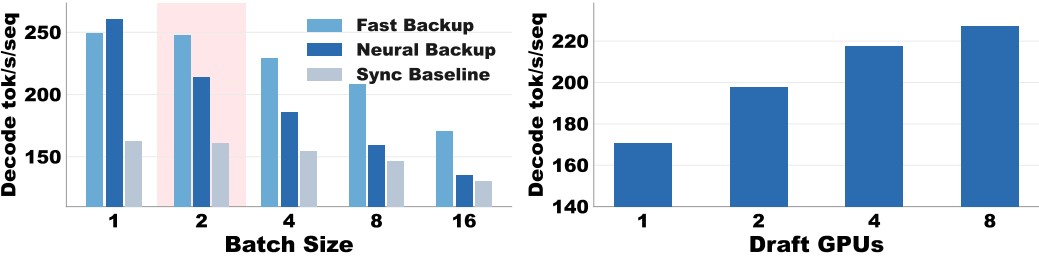

Figure 6: (Left) Optimal backup speculator (fast vs neural) depends on batch size, as Theorem 17 predicts (temperature 0). (Right) We forecast how decoding speed improves with draft compute (and thus, fan-out) for batch size 16 with the fast backup speculator. At large batch sizes we become compute bound, speedups more GPUs to drafting.

---

[2]Note that we can easily improve upon this random token strategy by using extremely fast non-neural speculators, like those based on n-grams (Liu et al., 2024; Oliaro et al., 2024).

## 5 END-TO-END EVALUATION

In Figure 7 we compare the end-to-end performance of SAGUARO to key baselines, including our own implementation of ordinary speculative decoding, as well as the strongest open source inference engines (vLLM/SGLang).[3] We also include EAGLE-3 when supported by open source engines.[4] We attain almost a 5x speedup on average compared to our autoregressive baseline, and are on average 30% faster than the strongest speculative decoding baselines, establishing a new state of the art.

Though speculative decoding algorithms are designed to trade off compute for lower latency, we show in Figure 7 (right) that SSD pushes the Pareto frontier across both latency and throughput, with the biggest gains at lower batch sizes. This means that SSD does not just improve the best possible decoding speed, *but is also more compute efficient per device.*

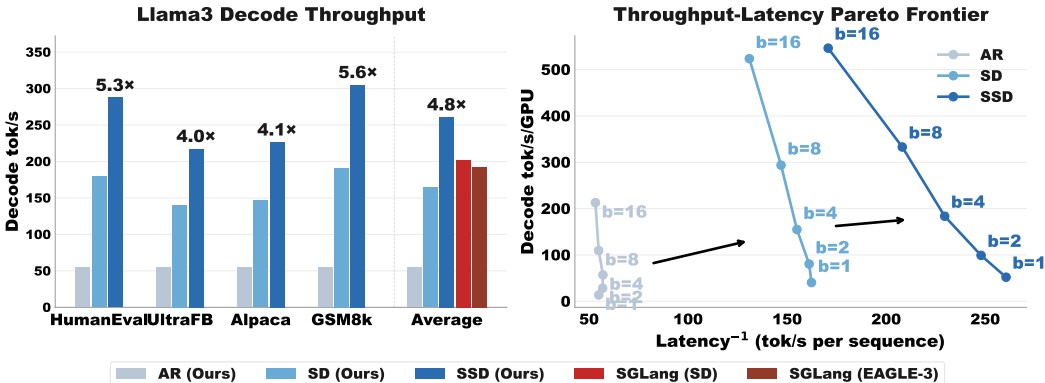

Figure 7: (Left) End-to-end decoding speed comparison of SSD compared to SD and standard autoregressive decoding on Llama-3.1-Instruct 70B across four datasets. (Right) SSD improves the throughput-latency Pareto frontier.

## 6 CONCLUSION AND LIMITATIONS

Speculative decoding trades compute for latency, but drafting and verification remain synchronous. We introduce speculative speculative decoding (SSD), which parallelizes even this sequential dependence. We derive performance bounds, study each component of the design space, and distill our findings into SAGUARO, an optimized SSD algorithm which is on average 30% faster than the strongest speculative decoding baselines and up to 5x faster than autoregressive generation across datasets and model families.

Speculative decoding focuses on reducing latency, and is generally ineffective for throughput-bound workloads—large-scale RL, offline data generation—since it adds verification workload to an already compute-bound process. Nonetheless, we show that SSD is able to push the latency-throughput Pareto frontier, leading to gains in both latency *and* throughput, even after taking into account the additional hardware used.

Much remains open. SSD composes naturally with EAGLE and token-tree speculation (Appendix E); the joint design and tradeoff space is largely unexplored. Latency can be further reduced by scaling the number of draft devices and thus the speculation cache, though the returns will eventually diminish. Finally, sharing speculation endpoints across several target model deployments at the cluster level—similar to prefill-decode disaggregation (Zhong et al., 2024)—is another natural direction.

---

[3]We find SGLang usually outperforms vLLM, so we use it as the main baseline throughout.
[4]In our experiments Llama-3.2-1B outperforms Eagle-3 as a draft model for Llama-3.1-70B in SGLang.

## ACKNOWLEDGEMENTS

We gratefully acknowledge the support of the Schmidt Sciences AI2050 fellowship, the Google ML and Systems Junior Faculty Awards, and the Google Research Scholar program. We thank Together AI for supporting this project with compute resources. Tanishq Kumar thanks Tatsunori Hashimoto and Luke Bailey for helpful comments and discussion.

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

## A    THEORETICAL RESULTS

Unless otherwise stated, we assume throughout that $T_p < 1$, so that $\max(T_p, 1) = 1$. This is because SSD has the primary speculator (draft with custom attention mask) overlapped with verification by construction, so that it finishes before verification is complete. The speculation lookahead is chosen to make this true in our implementation. We also assume that different sequences in a batch are unrelated, so that the probability of a cache hit is iid at $p_{\text{hit}}$.

### A.1    THEOREM 7 PROOF: MODELING THE SPEEDUP FROM SSD

We now prove Theorem 7. We include an (extended) copy of of the theorem here for reference:

**Theorem 7 (extended).**    *Let the primary and backup speculators take time $T_p$ and $T_b$ relative to the verifier. Let the expected number of generated tokens from the primary speculator be $E_{\text{hit}}$ and from the backup speculator be $E_{\text{miss}}$. The expected speedup of Algorithm 1 relative to autoregressive decoding is then:*

$$speedup \quad = \quad \frac{p_{\text{hit}} \cdot E_{\text{hit}} + (1 - p_{\text{hit}}) \cdot E_{\text{miss}}}{p_{\text{hit}} \cdot \max(1, T_p) + (1 - p_{\text{hit}}) \cdot (1 + T_b)}.$$

*The global cache hit rate $p_{\text{hit}}$ can be expressed in terms of the cache hit rates $p_{\text{hit,p}}$ and $p_{\text{hit,b}}$, corresponding to whether the last round's speculator was the primary one or the backup, respectively (assuming $|p_{\text{hit,p}} - p_{\text{hit,b}}| < 1$):*

$$p_{\text{hit}} \quad = \quad \frac{p_{\text{hit,b}}}{1 + p_{\text{hit,b}} - p_{\text{hit,p}}}. \tag{1}$$

*Proof.* We will prove this theorem in two parts:

- **Part 1**: We will first prove this speedup equation, *assuming we know the overall cache hit rate $p_{\text{hit}}$* (conditioned on choice of lookahead, cache topology, sampling algorithm, etc.).

- **Part 2**: We will then prove the functional form of $p_{\text{hit}}$, as a function of the cache hit rates $p_{\text{hit,p}}$ and $p_{\text{hit,b}}$ of the primary and backup speculators, respectively.

#### A.1.1    PART 1

In each iteration of SSD, the expected number of generated tokens is $E_{\text{hit}}$ if there is a cache hit, and $E_{\text{miss}}$ if there is not. Similarly, the latency is $\max(1, T_p)$ if there is a cache hit, and $1 + T_b$ if there is not—this is because backup speculation, which takes time $T_b$, begins only after verification (which we assume takes 1 unit of time) completes.

Therefore, it is clear that the expected number of generated tokens and the expected latency (relative to autoregressive decoding) in one iteration of SSD are:

$$\begin{aligned} E[\text{\# Generated tokens}] \quad &= \quad p_{\text{hit}} \cdot E_{\text{hit}} + (1 - p_{\text{hit}}) \cdot E_{\text{miss}} \\ E[\text{Latency}] \quad &= \quad p_{\text{hit}} \cdot \max(1, T_p) + (1 - p_{\text{hit}}) \cdot (1 + T_b) \end{aligned}$$

Thus, the expected speedup is:

$$\begin{aligned} speedup \quad &= \quad \frac{E[\text{\# Generated tokens}]}{E[\text{Latency}]} \\ &= \quad \frac{p_{\text{hit}} \cdot E_{\text{hit}} + (1 - p_{\text{hit}}) \cdot E_{\text{miss}}}{p_{\text{hit}} \cdot \max(1, T_p) + (1 - p_{\text{hit}}) \cdot (1 + T_b)}. \end{aligned}$$

This concludes part 1 of the proof.

### A.1.2   PART 2

We will now prove that the overall (unconditional) cache hit rate $p_{\text{hit}}$ is equal to:

$$p_{\text{hit}} \quad = \quad \frac{p_{\text{hit,b}}}{1 + p_{\text{hit,b}} - p_{\text{hit,p}}}.$$

where $p_{\text{hit,p}}$ and $p_{\text{hit,b}}$ are the cache hit rates conditioned on the prior iteration being speculated by the primary and backup speculators, respectively.

The challenge in deriving a closed-form solution for the overall cache hit rate $p_{\text{hit}}$ is that $p_{\text{hit}}$ at iteration $T$ of the SSD algorithm depends on whether there was a cache hit in the previous round (in which case the primary speculator was used) or not (in which case, the backup speculator was used).

In order to deal with this recursive property of $p_{\text{hit}}$, we first write $p_{\text{hit}}$ as a recursive equation, which considers the iteration number $t$ of the algorithm. We consider both the base case ($t = 0$), where for now we will assume we always use the non-neural spec, along with all rounds thereafter:

$$p_{\text{hit}}(0) \quad = \quad p_{\text{hit,p}}, \quad \text{because we use the primary speculator at } T = 0,$$

$$p_{\text{hit}}(T) \quad = \quad p_{\text{hit}}(T-1) \cdot p_{\text{hit,p}} + \left(1 - p_{\text{hit}}(T-1)\right) \cdot p_{\text{hit,b}}$$

We rearrange and unroll this recurrence:

$$
\begin{aligned}
p_{\text{hit}}(T) \quad &= \quad p_{\text{hit}}(T-1) \cdot \left(p_{\text{hit,p}} - p_{\text{hit,b}}\right) + p_{\text{hit,b}} \\
&= \quad p_{\text{hit}}(T-1) \cdot r + p_{\text{hit,b}}, \quad \text{letting } r := p_{\text{hit,p}} - p_{\text{hit,b}} \\
&= \quad \left(p_{\text{hit}}(T-2) \cdot r + p_{\text{hit,b}}\right) \cdot r + p_{\text{hit,b}} \\
&= \quad p_{\text{hit}}(T-2) \cdot r^2 + p_{\text{hit,b}} \cdot r + p_{\text{hit,b}} \\
&= \quad \left(p_{\text{hit}}(T-3) \cdot r + p_{\text{hit,b}}\right) \cdot r^2 + p_{\text{hit,b}} \cdot r + p_{\text{hit,b}} \\
&= \quad p_{\text{hit}}(T-3) \cdot r^3 + p_{\text{hit,b}} \cdot r^2 + p_{\text{hit,b}} \cdot r + p_{\text{hit,b}} \\
&= \quad \ldots \\
&= \quad p_{\text{hit}}(0) \cdot r^T + p_{\text{hit,b}} \sum_{t=0}^{T-1} r^t \\
&= \quad p_{\text{hit}}(0) \cdot r^T + p_{\text{hit,b}} \frac{1 - r^T}{1 - r}
\end{aligned}
$$

Using the stated assumption that $|r| := |p_{\text{hit,p}} - p_{\text{hit,b}}| < 1$, we can see that the first term above converges to 0, and the second term converges to $\frac{p_{\text{hit,b}}}{1-r} = \frac{p_{\text{hit,b}}}{1 + p_{\text{hit,b}} - p_{\text{hit,p}}}$, as expected.

This concludes the proof.

$\square$

### A.1.3   PROVING THE TWO COROLLARIES

We now prove the two corollaries of Theorem 7, copied below:

**Corollary 8. *(Strictly Faster Than SD)*** *Suppose we run SD with a given speculator $\mathcal{M}$. Running SSD with primary and backup both set to $\mathcal{M}$ does no worse than SD (strictly better if $p_{\text{hit}}, T_{\text{SD}} > 0$).*

*Proof.* Let $E_{\text{hit}} = E_{\text{miss}} = E_{\text{SD}}$, and $T_p = T_b = T_{\text{SD}}$. Then

$$
\begin{aligned}
speedup \quad &= \quad \frac{p_{\text{hit}} \cdot E_{\text{hit}} + (1 - p_{\text{hit}}) \cdot E_{\text{miss}}}{p_{\text{hit}} \cdot \max(1, T_p) + (1 - p_{\text{hit}}) \cdot (1 + T_b)} \\
&= \quad \frac{p_{\text{hit}} \cdot E_{\text{SD}} + (1 - p_{\text{hit}}) \cdot E_{\text{SD}}}{p_{\text{hit}} \cdot \max(1, T_{\text{SD}}) + (1 - p_{\text{hit}}) \cdot (1 + T_{\text{SD}})} \\
&= \quad \frac{E_{\text{SD}}}{p_{\text{hit}} \cdot \max(1, T_{\text{SD}}) + (1 - p_{\text{hit}}) \cdot (1 + T_{\text{SD}})}
\end{aligned}
$$

Recall that the speedup from SD is $E_{\text{SD}}/(1 + T_{\text{SD}})$. This final term is strictly greater than the SD speedup if $p_{\text{hit}} > 0$ and $T_{\text{SD}} > 0$, because in this case $\max(1, T_{\text{SD}}) < 1 + T_{\text{SD}}$. If $p_{\text{hit}} = 0$ or $T_{\text{SD}} = 0$, then the SSD speed is equal to the SD speed.

$\square$

**Corollary 9.** *(Speedup sandwich)* *Suppose we choose a primary speculator for which drafting completes before verification ($T_p < 1$), and a fast backup speculator ($T_b = 0$). Then if $T_{\text{SD}}, E_{\text{SD}}$ represent the latency and expected number of generated tokens from a draft model in SD, then the SSD speedup over SD can be bounded by:*

$$\left(1 + T_{\text{SD}}\right) \cdot \frac{E_{\text{hit}}}{E_{\text{SD}}} \cdot p_{\text{hit}} \leq \frac{speedup_{\text{SSD}}}{speedup_{\text{SD}}} \leq \left(1 + T_{\text{SD}}\right) \cdot \frac{E_{\text{hit}}}{E_{\text{SD}}}.$$

*Proof.* By assumption, $T_p < 1$ and $T_b = 0$. So the SSD speedup is:

$$speedup_{\text{SSD}} = \frac{p_{\text{hit}} \cdot E_{\text{hit}} + (1 - p_{\text{hit}}) \cdot E_{\text{miss}}}{p_{\text{hit}} \cdot \max(1, T_p) + (1 - p_{\text{hit}}) \cdot (1 + T_b)}$$

$$= p_{\text{hit}} \cdot E_{\text{hit}} + (1 - p_{\text{hit}}) \cdot E_{\text{miss}}$$

Recall the SD speedup equation is:

$$speedup_{\text{SD}} = \frac{E_{\text{SD}}}{1 + T_{\text{SD}}}.$$

So,

$$\frac{speedup_{\text{SSD}}}{speedup_{\text{SD}}} = \frac{p_{\text{hit}} \cdot E_{\text{hit}} + (1 - p_{\text{hit}}) \cdot E_{\text{miss}}}{E_{\text{SD}}/(1 + T_{\text{SD}})}.$$

Because $p_{\text{hit}} \leq 1$, we get the upper bound (assuming $E_{\text{hit}} \geq E_{\text{miss}}$):

$$\frac{speedup_{\text{SSD}}}{speedup_{\text{SD}}} = \left(1 + T_{\text{SD}}\right) \cdot \frac{E_{\text{hit}}}{E_{\text{SD}}}.$$

Because $E_{\text{miss}} \geq 1$ (bonus token is always generated), we get the lower-bound:

$$\frac{speedup_{\text{SSD}}}{speedup_{\text{SD}}} = \frac{p_{\text{hit}} \cdot E_{\text{hit}} + (1 - p_{\text{hit}}) \cdot E_{\text{miss}}}{E_{\text{SD}}/(1 + T_{\text{SD}})}$$

$$\geq \left(1 + T_{\text{SD}}\right) \cdot \frac{E_{\text{hit}}}{E_{\text{SD}}} \cdot p_{\text{hit}}.$$

$\square$

## A.2 THEOREM 12 PROOF: OPTIMIZING SAGUARO CACHE TOPOLOGY

To prove Theorem 12, we will first prove Theorem 18, which is a more general version of the theorem. Then, Theorem 12 will be an easy corollary of this more general theorem.

**Theorem 18.** *The choice of $F_k^p$ (and equivalently, $F_k^b$) values that maximizes the speedup of* SAGUARO *under the constraint $\sum_{k=0}^{K} F_k^p \leq B$ (where $K$ is the speculative lookahead), is:*

$$\sum_{k=0}^{K} F_k^p = B,$$

$$\frac{\partial p_{\text{hit,p}}^k}{\partial F}(F_k) = a_p^{-k} \cdot \frac{\partial p_{\text{hit,p}}^0}{\partial F}(F_0) \quad \forall k < K, \text{ and}$$

$$\frac{\partial p_{\text{hit,p}}^{K,all}}{\partial F}(F_K) = (1 - a_p) \cdot a_p^{-K} \cdot \frac{\partial p_{\text{hit,p}}^0}{\partial F}(F_0).$$

*Proof.* To understand how to optimize our cache topology, we first rewrite the speedup equations from Section 3.2, now making explicit how the speedup depends on the $F_k^p$ and $F_k^b$ values:

$$
\begin{aligned}
speedup\Big(\{F_k^p\}_{k=0}^K), \{F_k^b\}_{k=0}^K\Big)\ &=\ p_{\text{hit}}\Big(\{F_k^p\}, \{F_k^b\}\Big) \cdot E_{\text{hit}} \\
&\quad +\ \Big(1 - p_{\text{hit}}\Big(\{F_k^p\}, \{F_k^b\}\Big)\Big) \cdot E_{\text{miss}}, \quad \text{where} \\
p_{\text{hit}}\Big(\{F_k^p\}, \{F_k^b\}\Big)\ &=\ \frac{p_{\text{hit,b}}(\{F_k^b\})}{1 + p_{\text{hit,b}}(\{F_k^b\}) - p_{\text{hit,p}}(\{F_k^p\})}.
\end{aligned}
$$

We can express $p_{\text{hit,p}}$ more precisely in terms of:

- The acceptance rate $a_p$ of the primary speculator, and

- The functions $p_{\text{hit,p}}^k(F_k)$ and $p_{\text{hit,p}}^{K,all}(F_k)$: These functions describe the probability of a cache hit conditioned on (1) the last round's speculator was the primary one, (2) $k$ tokens were accepted, (3) whether all of the tokens were accepted ($p_{\text{hit,p}}^{K,all}$) or not ($p_{\text{hit,p}}^k$), and (4) $F_k$ verification outcomes were prepared for in case $k$ tokens were accepted. Note that this can be estimated empirically by comparing the draft and target probability distributions for a set of sequences in a calibration set.

We can do the same for $p_{\text{hit,b}}$, the cache hit rate when the last round's speculator was the backup.

$$
p_{\text{hit,p}}(\{F_k^p\})\ =\ a_p^K \cdot p_{\text{hit,p}}^{K,all}(F_K) + \sum_{k=0}^{K-1} a_p^k(1 - a_p) \cdot p_{\text{hit,p}}^k(F_k), \quad and
$$

These equations are a direct result of the fact that the chance of accepting exactly $k$ tokens is $a_p^k(1 - a_p)$ for $k < K$, and $a_p^K$ for $k = K$, when the acceptance rate is $a_p$.

Given these equations, we can now optimize the cache topology (i.e., the choice of $F_k$ values) to maximize speedup. We notice that the speedup is monotonically increasing in $p_{\text{hit}}$, and that $p_{\text{hit}}$ is monotonically increasing in both $p_{\text{hit,p}}$ and $p_{\text{hit,b}}$ (easy to see by taking first derivatives of $p_{\text{hit}}$, and seeing they are always positive). Thus, we must simply maximize $p_{\text{hit,p}}(\{F_k^p\})$ and $p_{\text{hit,b}}(\{F_k^b\})$ under the constraints that $\sum_{k=0}^K F_k^p \leq B$ and $\sum_{k=0}^K F_k^b \leq B$. We do this below.

### A.2.1 MAXIMIZING $p_{\text{hit,p}}(\{F_k^p\})$ AND $p_{\text{hit,b}}(\{F_k^b\})$

As discussed, we want to maximize the probability of a cache hit under the budget constraint $\sum_{k=0}^K F_k \leq B$. We do this for $p_{\text{hit,p}}$, and the proof is identical for $p_{\text{hit,b}}$.

$$
\max_{\sum_k F_k \leq B} p_{\text{hit,p}}(K, F) = \max_{\sum_k F_k \leq B} a_p^K \cdot p_{\text{hit,p}}^{K,all}(F_K) + \sum_{k=0}^{K-1} a_p^k(1 - a_p) \cdot p_{\text{hit,p}}^k(F_k)
$$

We will solve this maximization problem with Lagrange multipliers.

$$
\mathcal{L}(F_0, \ldots, F_k, \lambda) = a_p^K \cdot p_{\text{hit,p}}^{K,all}(F_K) + \sum_{k=0}^{K-1} a_p^k(1 - a_p) \cdot p_{\text{hit,p}}^k(F_k) + \lambda \cdot \left(\sum_{k=0}^K F_k - B\right)
$$

Now, we will take the derivative with respect to all the variables, and set it to zero.

$$\frac{\partial \mathcal{L}}{\partial F_k} = a_p^k(1 - a_p) \cdot \frac{\partial}{\partial F_k} p_{\text{hit,p}}^k(F_k) + \lambda = 0 \quad \text{for } k < K$$

$$\frac{\partial \mathcal{L}}{\partial F_K} = a_p^K \frac{\partial}{\partial F_k} p_{\text{hit,p}}^{K,all}(F_K) + \lambda = 0$$

$$\frac{\partial \mathcal{L}}{\partial \lambda} = \sum_{k=0}^{K} F_k - B = 0$$

We notice that for all $k$, $\frac{\partial \mathcal{L}}{\partial F_k}$ are equal to one another (all equal to $-\lambda$). Thus, we see that:

$$(1 - a_p)\frac{\partial}{\partial F_k} p_{\text{hit,p}}^0(F_0) = a_p(1 - a_p)\frac{\partial}{\partial F_k} p_{\text{hit,p}}^1(F_1) = \ldots$$

$$\ldots = a_p^{K-1}(1 - a_p)\frac{\partial}{\partial F_k} p_{\text{hit,p}}^{K-1}(F_{K-1}) = a_p^K \frac{\partial}{\partial F_k} p_{\text{hit,p}}^{K,all}(F_K)$$

$$\Rightarrow \frac{\partial}{\partial F_k} p_{\text{hit,p}}^k(F_k) = a_p^{-k}\frac{\partial}{\partial F_k} p_{\text{hit,p}}^0(F_0) \quad \forall k < K, \text{ and}$$

$$\frac{\partial}{\partial F_k} p_{\text{hit,p}}^{K,all}(F_K) = (1 - a_p)a_p^{-K}\frac{\partial}{\partial F_k} p_{\text{hit,p}}^0(F_0)$$

This gives the desired result.

$\square$

We now use this result to prove (an extended version of) Theorem 12, which is a special case of the above theorem in the case where the speculators have $r$ power-law cache hit rates.

**Theorem 12. (SAGUARO *Cache Shape: Geometric Fan-Out*)** *The optimal choice of $F_k^p$ (equivalently, $F_k^b$) values for $k \in [0, K]$ for SAGUARO under the constraint $\sum_{k=0}^{K} F_k^p \leq B$, and under the assumption that the speculator has acceptance rate $a_p$ and a $r$ power-law cache hit rate, follows a geometric series (for $k < K$):*

$$F_k = F_0 \cdot a_p^{k/(1+r)} \quad \forall k < K, \text{ and}$$

$$F_K = F_0 \cdot a_p^{K/(1+r)} \cdot (1 - a_p)^{-1/(1+r)},$$

*where $F_0$ can be chosen as follows so that $\sum_{k=0}^{K} F_k^p = B$.*

$$F_0 = \frac{B}{a_p^{K/(1+r)} \cdot (1 - a_p)^{-1/(1+r)} + \left(1 - a_p^{K/(1+r)}\right)/\left(1 - a_p^{1/(1+r)}\right)}$$

*An equivalent result holds for $F_k^b$.*

*Proof.* Now, substituting $p_{\text{hit,p}}^k(F) = 1 - F^{-r}$ (and thus $\frac{\partial}{\partial F_k} p_{\text{hit,p}}^k(F) = r \cdot F^{-r-1}$), we get:

$$\Rightarrow r \cdot F_k^{-r-1} = a_p^{-k} r \cdot F_0^{-r-1} \quad \forall k < K, \text{ and}$$

$$r \cdot F_K^{-r-1} = (1 - a_p) \cdot a_p^{-K} \cdot r \cdot F_0^{-r-1}.$$

$$\Rightarrow F_k = F_0 \cdot a_p^{k/(1+r)} \quad \forall k < K, \text{ and}$$

$$F_K = F_0^{(1+r)/(1+r)} \cdot a_p^{K/(1+r)} \cdot (1 - a_p)^{-1/(1+r)} \cdot (r/r)^{-1/(1+r)}.$$

To solve for the exact sequence of $F_k$ fan-out values, we plug the above values into the budget equation:

$$B = F_0 \cdot a_p^{K/(1+r)} \cdot (1 - a_p)^{-1/(1+r)} + \sum_{k=0}^{K-1} F_0 \cdot a_p^{k/(1+r)}$$

Solving for $F_0$ gives:

$$F_0 \;=\; \frac{B}{a_p^{K/(1+r)} \cdot (1 - a_p)^{-1/(1+r)} + \sum_{k=0}^{K-1} a_p^{k/(1+r)}}$$

We can simplify this further using the equation for the sum of the geometric series:

$$\sum_{k=0}^{K-1} a_p^{k/(1+r)} = \sum_{k=0}^{K-1} c^k = \frac{1 - c^K}{1 - c}, \quad \text{for } c = a_p^{1/(1+r)}$$

Plugging this in gives the desired result.

$$F_0 \;=\; \frac{B}{a_p^{K/(1+r)} \cdot (1 - a_p)^{-1/(1+r)} + \left(1 - a_p^{K/(1+r)}\right) / \left(1 - a_p^{1/(1+r)}\right)}$$

$\square$

### A.3 THEOREM 15 PROOF: OPTIMIZING SAGUARO SAMPLING ALGORITHM

**Theorem 15.** *For fan-out $F$, and draft logits $z$, the cache hit rate $p_{\mathrm{hit}}$ of the* SAGUARO *sampling algorithm increases as $C \to 0$.*

*Proof.* Fix fan-out $F$ and draft logits $z$. Let $S := \mathrm{top}_F(z)$ denote the $F$ cached tokens (where $S$ depends only on $z$, not on $C$). Write $w_t := \exp(z_t) > 0$ and define

$$A := \sum_{t \in S} w_t, \qquad B := \sum_{t \notin S} w_t, \qquad Z(C) := CA + B.$$

Then the SAGUARO sampling scheme induces the draft distribution

$$p_C(t) \;=\; \begin{cases} \dfrac{Cw_t}{Z(C)} & t \in S, \\[2mm] \dfrac{w_t}{Z(C)} & t \notin S. \end{cases}$$

For $t \in S$,

$$\frac{d}{dC} p_C(t) \;=\; \frac{w_t B}{(CA + B)^2} \;\geq\; 0,$$

and for $t \notin S$,

$$\frac{d}{dC} p_C(t) \;=\; -\frac{w_t A}{(CA + B)^2} \;\leq\; 0.$$

Thus increasing $C$ increases $p_C(t)$ on cached tokens and decreases $p_C(t)$ off the cache.

Let $q(t) := p_{\mathrm{target}}(t)$ and define the (unnormalized) residual mass

$$u_C(t) := \max\big(q(t) - p_C(t), \, 0\big).$$

Since $u_C(t)$ is a nonincreasing function of $p_C(t)$, the above monotonicity gives that for $t \in S$, $u_C(t)$ is nonincreasing in $C$; for $t \notin S$, $u_C(t)$ is nondecreasing in $C$. Then define

$$\mathrm{in}(C) := \sum_{t \in S} u_C(t), \qquad \mathrm{out}(C) := \sum_{t \notin S} u_C(t).$$

Then $\mathrm{in}(C)$ is nonincreasing in $C$ and $\mathrm{out}(C)$ is nondecreasing in $C$.

For any verification case in which the bonus token is sampled from the residual distribution (i.e., when $k < K$ in SD/SSD), the probability that the sampled bonus token lies in the cache $S$ is

$$p_{\mathrm{hit}}(C) \;=\; \frac{\mathrm{in}(C)}{\mathrm{in}(C) + \mathrm{out}(C)},$$

with the convention that if $\mathrm{in}(C) + \mathrm{out}(C) = 0$ (zero residual mass), then $p_{\mathrm{hit}}(C) = 1$. Then since the map $(a, b) \mapsto a/(a + b)$ is nondecreasing in $a$ and nonincreasing in $b$ for $a, b \geq 0$, and since $\mathrm{in}(C)$ decreases while $\mathrm{out}(C)$ increases with $C$, it follows that $p_{\mathrm{hit}}(C)$ is nonincreasing in $C$, concluding the proof. $\square$

### A.3.1 Construction: Saguaro Sampling Can Give Speedups

**Theorem 19.** *There exist target and draft distributions $p_t$ and $p_d$ where applying* Saguaro *sampling to the draft distribution $p_d$ using some $C \in (0,1)$ leads to strictly faster end-to-end speeds than sampling from $p_d$ directly.*

*Proof.* We now construct target and draft distributions $p_t$ and $p_d$ over a vocabulary of 4 tokens that satisfy the above property:

**Construction 1.** We define $p_t$ and $p_d$ as follows:
$$p_t := (0.48,\ 0.48,\ 0.02,\ 0.02),$$
$$p_d := (0.49,\ 0.49,\ 0.01,\ 0.01).$$

We will now show that applying Saguaro sampling to $p_d$ with $C = 47/147$ leads to a draft distribution $p'_d$ with strictly faster end-to-end speeds than $p_d$. We prove this by showing that $p'_d$ has the same acceptance rate as $p_d$, but has higher cache hit rates. Applying Theorem 7 shows that $p'_d$ gives faster end-to-end speedups than $p_d$.

**Part 1. Acceptance rate is unchanged.** It is easy to see that applying Saguaro sampling to $p_d$ with $C = 47/147$ and $F = 2$ produces
$$p'_d = (0.47,\ 0.47,\ 0.03,\ 0.03).$$
We can then see that the acceptance rate for both $p_d$ and $p'_d$ is equal to 0.98.
$$a(p_t,\ p_d) = 1 - \|p_d - p_t\|_1 = 1 - 0.04/2 = 0.98$$
$$a(p_t,\ p'_d) = 1 - \|p'_d - p_t\|_1 = 1 - 0.04/2 = 0.98$$

**Part 2. Cache hit rate strictly increases.** We now show $p'_d$ has a strictly higher cache hit rate than $p_d$. We begin by computing the residual distributions $r$ and $r'$ for $p_d$ and $p'_d$, respectively:
$$r = (0,\ 0,\ 0.5,\ 0.5)$$
$$r' = (0.5,\ 0.5,\ 0,\ 0)$$

*Cache hit rate for $p_d$:* To compute the cache hit rate for $p_d$, we need to consider the case where a sample from $p_d$ was rejected. If token 3 or 4 was sampled from $p_d$ it would always be accepted, so a rejected implies that either tokens 1 or 2 was sampled. Assume without loss of generality that it was token 1 that was sampled and then rejected. In this case, the speculation cache would contain tokens 2 and 3 (or equivalently, 2 and 4). The residual distribution for $p_d$ is $r = (0,\ 0,\ 0.5,\ 0.5)$, so with 50% probability token 3 (which is in the speculation cache) would be sampled. So the cache hit rate is 50%.[5]

*Cache hit rate for $p'_d$:* Just like we did above, to compute the cache hit rate for $p_d$, we need to consider the case where a sample from $p'_d$ was rejected. If token 1 or 2 was sampled from $p'_d$ it would always be accepted, so a rejected implies that either tokens 3 or 4 was sampled. Assume without loss of generality that it was token 3 that was sampled and then rejected. In this case, the speculation cache would contain tokens 1 and 2. The residual distribution for $p_d$ is $r = (0.5,\ 0.5,\ 0,\ 0)$, so with 100% probability either token 1 or 2 would be sampled (which are both in the cache) would be sampled. So the cache hit rate is 100%.

This concludes the proof, since we have shown that Saguaro sampling can transform $p_d$ into $p'_d$, which has the same acceptance rate as $p_d$, but has strictly higher cache hit rate.

### A.4 Corollary 16 and Theorem 17 Proofs: Optimizing Saguaro Fallback Strategy

We first prove corollary 16, and then Theorem 17, both copied below for reference:

---

[5] Note that this is the cache hit rate $p_{hit,p}$ conditioned that the prior speculation was sampled from the primary speculator. It is sufficient to only consider this cache hit rate, and not $p_{hit,b}$, since this quantity would be the same for both $p_d$ and $p'_d$. Thus, using Equation 1, we can see that showing that the conditional acceptance rate $p_{hit,p}$ is higher for $p'_d$ than $p_d$ is sufficient to show the unconditional acceptance rate $p_{hit}$ is also higher.

**Corollary 16.** *At batch size b, the expected speedup from SSD is equal to:*

$$speedup \quad = \quad \frac{p_{\text{hit}} \cdot E_{\text{hit}} + (1 - p_{\text{hit}}) \cdot E_{\text{miss}}}{p_{\text{hit}}^b \cdot \max(1, T_p) + (1 - p_{\text{hit}}^b) \cdot (1 + T_b)}, \quad \textit{which approaches}$$

$$\frac{p_{\text{hit}} \cdot E_{\text{hit}} + (1 - p_{\text{hit}}) \cdot E_{\text{miss}}}{1 + T_b} \quad \textit{as } b \to \infty.$$

*Proof.* For each element of the batch, if it gets a cache hit it will generate $E_{\text{hit}}$ tokens, and otherwise $E_{\text{miss}}$. The latency of an element of the batch, however, depends on whether any element of the batch had a cache miss. If so, the latency for the entire batch is $1 + T_b$. Otherwise, if every element of the cache had a hit, the latency is $\max(1, T_p)$. Thus, we can see that:

$$E[\text{\# Generated tokens}] \quad = \quad p_{\text{hit}} \cdot E_{\text{hit}} + (1 - p_{\text{hit}}) \cdot E_{\text{miss}}$$
$$E[\text{Latency}] \quad = \quad p_{\text{hit}}^b \cdot \max(1, T_p) + (1 - p_{\text{hit}}^b) \cdot (1 + T_b)$$

Thus, the expected speedup is:

$$speedup \quad = \quad \frac{E[\text{\# Generated tokens}]}{E[\text{Latency}]}$$

$$= \quad \frac{p_{\text{hit}} \cdot E_{\text{hit}} + (1 - p_{\text{hit}}) \cdot E_{\text{miss}}}{p_{\text{hit}}^b \cdot \max(1, T_p) + (1 - p_{\text{hit}}^b) \cdot (1 + T_b)}.$$

Noticing that $p_{\text{hit}}^b \to 0$ as $b$ grows concludes the proof.

$\square$

**Theorem 17.** *The optimal cache miss strategy, conditioned on only being able to choose between a high-quality slow speculator (primary), and a lower-quality speculator with negligible latency (backup), is to use the primary speculator for batch sizes $b < b^*$, and the backup speculator otherwise. The value of $b^*$ is given by:*

$$b^* \quad = \quad \frac{1}{\log(p_{\text{hit}})} \cdot \log\left(\left(1 + \frac{1}{T_p} - \frac{E_{\text{hit}}}{T_p \cdot p_{\text{hit}} \cdot E_{\text{hit}} + T_p \cdot (1 - p_{\text{hit}}) \cdot E_{\text{miss}}}\right)\right)$$

*Proof.* Using the slow primary speculator as the backup speculator gives expected speedup:

$$speedup_{\text{slow\_backup}} \quad = \quad \frac{p_{\text{hit}} \cdot E_{\text{hit}} + (1 - p_{\text{hit}}) \cdot E_{\text{hit}}}{p_{\text{hit}}^b \cdot \max(1, T_p) + (1 - p_{\text{hit}}^b) \cdot (1 + T_p)}.$$

$$= \quad \frac{E_{\text{hit}}}{p_{\text{hit}}^b + (1 - p_{\text{hit}}^b) \cdot (1 + T_p)}.$$

$$= \quad \frac{E_{\text{hit}}}{1 + T_p - T_p \cdot p_{\text{hit}}^b}.$$

Using the fast backup speculator (with $T_b = 0$) as the backup speculator gives expected speedup:

$$speedup_{\text{fast\_backup}} \quad = \quad \frac{p_{\text{hit}} \cdot E_{\text{hit}} + (1 - p_{\text{hit}}) \cdot E_{\text{miss}}}{p_{\text{hit}}^b \cdot \max(1, T_p) + (1 - p_{\text{hit}}^b) \cdot (1 + T_b)}.$$

$$= \quad \frac{p_{\text{hit}} \cdot E_{\text{hit}} + (1 - p_{\text{hit}}) \cdot E_{\text{miss}}}{p_{\text{hit}}^b \cdot \max(1, T_p) + (1 - p_{\text{hit}}^b)}.$$

$$= \quad p_{\text{hit}} \cdot E_{\text{hit}} + (1 - p_{\text{hit}}) \cdot E_{\text{miss}}.$$

We set these equations equal to each other and solve for $b$, which gives the equation in the Theorem statement.

$$
\begin{aligned}
p_{\text{hit}} \cdot E_{\text{hit}} + (1 - p_{\text{hit}}) \cdot E_{\text{miss}} &= \frac{E_{\text{hit}}}{1 + T_p - T_p \cdot p_{\text{hit}}^b} \\
1 + T_p - T_p \cdot p_{\text{hit}}^b &= \frac{E_{\text{hit}}}{p_{\text{hit}} \cdot E_{\text{hit}} + (1 - p_{\text{hit}}) \cdot E_{\text{miss}}} \\
p_{\text{hit}}^b &= \frac{1}{T_p}\left(1 + T_p - \frac{E_{\text{hit}}}{p_{\text{hit}} \cdot E_{\text{hit}} + (1 - p_{\text{hit}}) \cdot E_{\text{miss}}}\right) \\
b^* &= \frac{1}{\log(p_{\text{hit}})} \cdot \log\left(\left(1 + \frac{1}{T_p} - \frac{E_{\text{hit}}}{T_p \cdot p_{\text{hit}} \cdot E_{\text{hit}} + T_p \cdot (1 - p_{\text{hit}}) \cdot E_{\text{miss}}}\right)\right).
\end{aligned}
$$

Now we must simply show that for batch sizes $b < b^*$, it is better to use the slow backup speculator (a.k.a., the primary speculator), and for $b \geq b^*$ it is better to use the fast backup speculator.

We do this by seeing that the $speedup_{\text{slow\_backup}}$ is monotonically decreasing in the batch size $b$ (negative derivative with respect to $b$), whereas the $speedup_{\text{fast\_backup}}$ does not depend on $b$ (obvious from equation). This shows that if there is a value $b^*$ that makes these two speedups equal, then for $b < b^*$ the slow backup option gives a larger speedup, whereas for $b \geq b^*$ the fast backup option gives a larger speedup.

$$
\begin{aligned}
\frac{\partial speedup_{\text{slow\_backup}}}{db} &= \frac{\partial}{db}\left(\frac{E_{\text{hit}}}{1 + T_p - T_p \cdot p_{\text{hit}}^b}\right) \\
&= \frac{E_{\text{hit}} \cdot T_p \cdot p_{\text{hit}}^b \cdot \log(p_{\text{hit}})}{(1 + T_p - T_p \cdot p_{\text{hit}}^b)^2},
\end{aligned}
$$

which is clearly negative because $\log(p_{\text{hit}}) < 0$ and everything else is positive. This concludes the proof.

$\square$

# B  IMPLEMENTATION DETAILS

## B.1  SYSTEMS DESIGN

**Overall Design.** We implement SAGUARO in a custom inference engine written in PyTorch. Our implementation incorporates PagedAttention (Kwon et al., 2023), continuous batching (Yu et al., 2022), tensor parallelism, BF16 mixed precision, torch compilation, and CUDAGraphs.

The architecture is built around a target model split across 4 devices, and a small draft on a separate device. The two communicate through NCCL once per speculation round, wherein the target sends only the results of the previous round (number of accepted tokens, bonus token sampled), and the draft model uses this to key a lookup into its pre-prepared speculation cache, and immediately returns the tokens and logits corresponding to the pre-prepared speculation on a cache hit, and either random outputs or just-in-time speculation in the case of a cache miss. The key point is that the target model never sees the draft-side speculation cache: it is sent the speculated tokens for its verification outcome immediately after verification. In particular, no KV cache is ever transferred between the devices.

**Implementation Details.** The engine orchestrates from a coordinator process on the main GPU. A scheduler paired with a block manager handles prefill/decode scheduling and page-table bookkeeping. A ModelRunner on each target GPU prepares attention metadata and executes forward passes. In async mode, the draft model runs in a separate process on its own GPU. Target and draft communicate once per iteration via NCCL with fused payloads: the target sends cache keys (sequence ID, accepted-prefix length, recovery token), current sequence lengths, draft block tables for KV addressing, and per-row temperatures; the draft returns a cache-hit bitmap, $K$ speculative tokens per sequence, and $K$-step logits for acceptance.

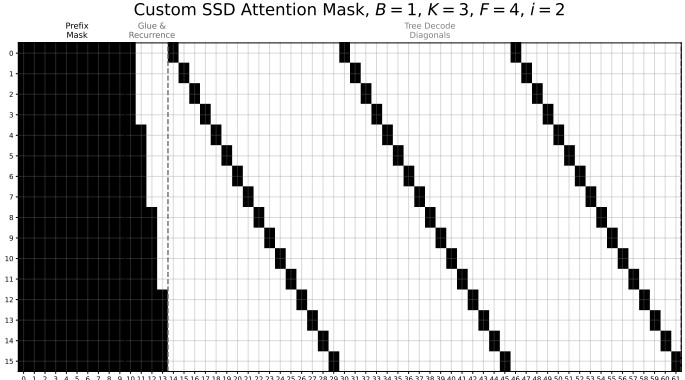

Figure 8: Custom attention mask for multi-query decoding of all $BF(K+1)$ verification branches in parallel. This mask is for $B = 1$, $K = 3$, uniform fan-out $F = 4$, at depth $i = 2$. Black indicates tokens that can be attended to. The left block shows attention to the verified prefix; diagonal bands show each branch attending only within its forking path.

While the target host maintains page-table bookkeeping for both models, the draft KV cache tensor resides on the draft device. The scheduler ensures the target has sufficient pages for $K + 1$ multi-query decoding steps, preempting sequences when lookahead reservations cannot be satisfied. After verification, both page tables are reconciled: completed pages are finalized (hashed for prefix caching), and any pages allocated beyond the accepted suffix are deallocated. This rollback requirement—undoing allocations for rejected tokens that wrote into pre-allocated pages—necessitates a host-side post-processing step after each verification.

**Design Decisions and Performance Engineering.** During draft speculation, all $F(K + 1)$ branches of each sequence are decoded in parallel using a custom sparse attention mask that allows each branch to attend to the verified trunk and its own forking path. We use FlashAttention kernels (Shah et al., 2024) where possible, falling back to FlashInfer (Ye et al., 2025) for multi-query decoding paths requiring custom masks. An example mask is shown in Figure 8.

Materializing these masks—which depend on prefix length, $B$, $K$, $F$, and step $i$—is substantial overhead. The sparse, non-coalesced memory access patterns in custom-mask attention kernels dominate our critical path, limiting how many steps we can profitably draft. Thus, most end-to-end speedup comes from hiding draft latency rather than increasing lookahead. Since accepted branches land in fragmented KV cache locations, we perform an extend (prefill) operation of the previous speculation into KV cache before each round of async decoding. This corresponds to the "Glue & Recurrence" column in Figure 8, enabling all $F$ forked branches to attend to the same prefix.

### B.2 EXPERIMENTAL DESIGN

**Datasets.** We take 128 prompts from each dataset and sample 512 decoding tokens for each. We use vanilla sampling (not top-p or top-k). We measure decoding throughput, excluding prefill. All experiments are done on a single node of NVIDIA H100s, though we only use TP=4 (AR, SD) or 5 (SSD) GPUs at once.

**Baselines.** We compare to the two most widely used open-source inference engines: vLLM (v0.16.0) and SGLang (v0.5.9), with autoregressive, standalone speculative decoding, and EAGLE-3. All baselines use tensor parallelism across 4×H100 80GB GPUs with greedy decoding, radix caching disabled, and a single concurrent request (batch size 1) unless otherwise specified. For speculative decoding, we use Llama-3.2-1B-Instruct as the draft model for Llama-3.1-70B-Instruct and Qwen3-0.6B for Qwen3-32B, proposing 5 draft tokens per step. We evaluate on 512 prompts drawn equally from HumanEval, Alpaca, GSM8K, and UltraFeedback, generating 512 tokens per prompt. Though SSD outperforms tree-based methods like EAGLE-3, it can be combined with these methods—see Appendix E. Our implementation of vanilla speculative decoding has speed comparable to that of vLLM and SGLang confirming it is a strong baseline.

## B.3 NUMERICAL RESULTS

| Model | Dataset | AR tok/s | SD tok/s | SSD tok/s | SSD/SD | SSD/AR |
|---|---|---|---|---|---|---|
| Llama-3.1-Instruct | | | | | | |
| 70B/1B | HumanEval | 54.7 | 176 | 283 | 1.60× | 5.17× |
| | Ultrafeedback | 54.7 | 138 | 215 | 1.55× | 3.93× |
| | Alpaca | 54.7 | 145 | 224 | 1.55× | 4.10× |
| | GSM8k | 54.7 | 188 | 301 | 1.60× | 5.50× |
| | Average | 54.7 | 161.8 | 255.8 | 1.58× | 4.68× |
| Qwen-3 | | | | | | |
| 32B/0.6B | HumanEval | 88.8 | 146 | 222 | 1.52× | 2.50× |
| | Ultrafeedback | 88.8 | 122 | 174 | 1.43× | 1.96× |
| | Alpaca | 88.8 | 127 | 185 | 1.47× | 2.08× |
| | GSM8k | 88.8 | 152 | 234 | 1.54× | 2.64× |
| | Average | 88.8 | 136.8 | 203.8 | 1.49× | 2.29× |

## B.4 BASELINE SWEEPS

Cells report decode throughput (tok/s); † indicates incompatible configs due to known bugs in the library version used (SGLang 0.5.9 / vLLM 0.16.0). The best configuration per method is **bolded**. We sweep speculative lookahead and tree size for both vanilla SD and EAGLE-3 for SGLang, and just lookahead for vLLM since tree verification codepaths are known to be incomplete.

| SGLang EAGLE-3 (steps × topk) | | | | | |
|---|---|---|---|---|---|
| steps \ topk | 1 | 2 | 3 | 4 | 5 |
| 2 | 122.8 | 124.7 | 127.3 | † | † |
| 4 | 157.1 | 157.8 | 164.9 | † | † |
| 6 | 171.3 | 178.9 | 183.0 | † | † |
| 8 | 172.4 | 183.9 | **192.8** | 190.2 | † |
| 10 | 168.7 | 183.0 | 192.5 | † | 186.9 |
| 12 | 164.8 | 178.7 | 186.3 | 187.6 | 185.4 |
| 14 | 158.4 | 173.8 | 180.3 | 182.9 | 176.9 |

| SGLang Standalone (steps × topk) | | | | | |
|---|---|---|---|---|---|
| steps \ topk | 1 | 2 | 3 | 4 | 5 |
| 2 | 123.0 | 123.9 | 125.4 | † | † |
| 4 | 159.4 | 161.2 | 164.9 | † | † |
| 6 | 176.7 | 182.3 | 186.7 | † | 164.9 |
| 8 | 182.1 | 192.3 | 197.7 | † | 163.1 |
| 10 | 181.5 | 195.7 | **201.9** | 178.0 | 160.3 |
| 12 | 180.4 | 195.2 | 199.0 | † | † |
| 14 | 175.7 | 193.6 | † | 171.5 | † |

Table 1: SGLang 0.5.9 Llama-3.1-70B/1B baseline sweeps (decode tok/s).

| Method | $K = 2$ | $K = 4$ | $K = 6$ | $K = 8$ | $K = 10$ | $K = 12$ |
|---|---|---|---|---|---|---|
| vLLM EAGLE-3 | 143.7 | 181.4 | **197.6** | 197.0 | 191.8 | 187.9 |
| vLLM Standalone | **136.0** | 117.9 | 100.9 | 89.1 | 78.5 | 70.5 |

Table 2: vLLM 0.16.0 Llama-3.1-70B/1B baseline sweeps over speculative lookahead $K$ (decode tok/s).

## C SSD OVERHEAD

Let $B$ be the target batch size, $K$ the speculation lookahead, and $F$ the (uniform) fan out factor for the draft model guessing verification bonus tokens.

**Compute.** Let $\hat{c}$ be the compute required for a draft forward pass, normalized by units of target FLOPs. Since the draft decodes $BK(K + 1)F$ tokens per round in SSD but $BK$ tokens per round in SD, we incur a factor of $\hat{c}(K + 1)F$ more FLOPs on the draft relative to ordinary speculative

decoding. Recall this is because each of the $K$ draft steps decodes $B(K + 1)F$ tokens in parallel using a custom attention mask.

In much the same way that SD uses more FLOPs to achieve lower latency (see (Leviathan et al., 2023), Section 3.4), SSD applies the same philosophy to use *even more FLOPs* to achieve *even lower latency* than even SD itself. In the SD setting, tokens speculated by the draft that were rejected by the target constitute wasted compute. In the SSD setting (in addition to the above), we have that entire *chains* decoded pre-emptively in parallel for anticipated verification outcomes constitute wasted compute. Thus, SSD introduces new tradeoffs between compute and latency that were not possible before.

**Memory.** The draft model must build up a speculation cache as it speculates asynchronously. This has possible verification outcomes as keys and the corresponding tokens/logits as values. Concretely, this means storing a tensor of $BK(K + 1)F$ tokens that are decoded, in the form of length-$K$ speculations stored for $B(K + 1)F$ possible verification outcomes. For each of these tokens, we also store logits of size $V$, so the speculation cache stores overall $O\left(BFK(K + 1)(V + 1)\right)$ bits, ending up around hundreds of megabytes in practice. Given this cache is refreshed every speculation round, this ends up small enough to be a non-issue in practice, as the HBM on modern GPUs is much larger.

**Communication.** The draft and target model synchronize once per speculation round. The target model sends the outcome of the previous round of speculation (the number of accepted tokens and recovery token for each sequence, $O(B)$ bits of information). The draft sends back the newly speculated tokens ("cache hits") as well as their logits (which the target will need for verification). This is $O(BKV)$ bits of information. All communications are device to device over NCCL via NVLink, which is fast enough that communications are not a bottleneck in practice.

## D    EXTENDED RELATED WORK

SSD is complementary to other lines of work that accelerate LLM inference through orthogonal means: hardware-aware algorithms like FlashAttention (Dao et al., 2022), KV-cache management and paging (Kwon et al., 2023; Zhang et al., 2023; Hooper et al., 2024), sparse or approximate attention (Beltagy et al., 2020; Zaheer et al., 2020), and multi-token or tree-based draft methods (Cai et al., 2024; Fu et al., 2024). SSD targets a different bottleneck—the sequential dependence between drafting and verification—and can in principle be combined with any of these.

## E    COMBINING SSD WITH SD VARIANTS

**EAGLE-3.** EAGLE-3 (Li et al., 2025b) improves the acceptance rate $\alpha$ by training the draft to condition on target activations extracted during the previous round's verification. When target activations are unavailable mid-speculation, the draft conditions on its own activations as a surrogate, with acceptance degrading slowly as more self-generated activations are used.

This is entirely compatible with SSD, with one nuance: since the SSD draft begins speculating round $T+1$ pre-emptively, it does not yet have target activations from round $T$ (unlike standard EAGLE-3, which waits for them). With lookahead $K$, the draft thus conditions on up to $2K$ self-generated tokens, lowering acceptance on the latter $K$. This can be mitigated by training the EAGLE-3 draft to preserve acceptance over longer periods of self-conditioning. This is depicted in Figure 9.

In our implementation, the target model sends the number of accepted tokens per sequence as well as the target activations at each relevant token position to the draft over NCCL for the draft model to condition on when speculating.

**Token-Tree Methods.** Token-tree SD methods (Miao et al., 2024; Chen et al., 2024; Li et al., 2024b; 2025b) can be combined naturally with SSD. SSD drafts a linear chain for many possible verification outcomes. To combine SSD with these token-tree approaches, one would simply pre-speculate (and verify) a token tree for each verification outcome instead of a token chain. In practice, this requires a more elaborate attention mask than that presented in Figure 8.

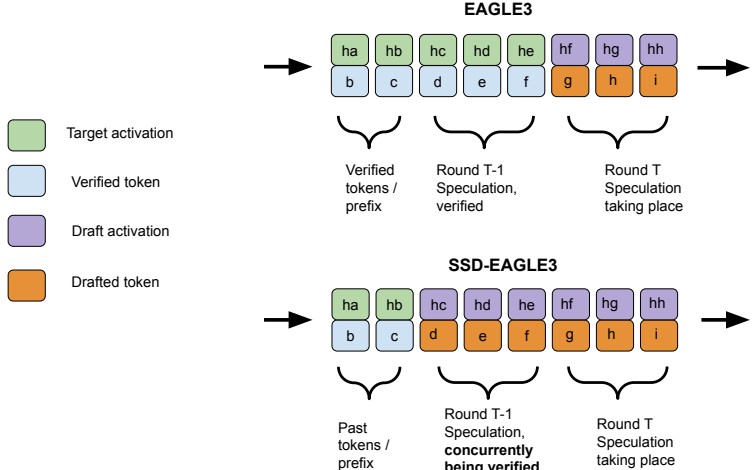

Figure 9: Comparison of activation conditioning in EAGLE-3 vs. SSD-EAGLE-3. In standard EAGLE-3 (bottom), the round $T$ draft conditions almost entirely on target activations extracted from the already-verified round $T-1$. In SSD-EAGLE-3 (top), round $T$ speculation begins *before* round $T-1$ verification completes, so the draft must substitute its own activations (light nodes) for the not-yet-available target activations. Conditioning on more draft activations can degrade quality of EAGLE speculations if the draft model is not trained to use draft self-conditioning effectively.

## F ADDITIONAL EXPERIMENTS

Here, we reproduce similar trends from the main text on the Qwen3 model family, demonstrating that our results and algorithms are model and dataset agnostic.

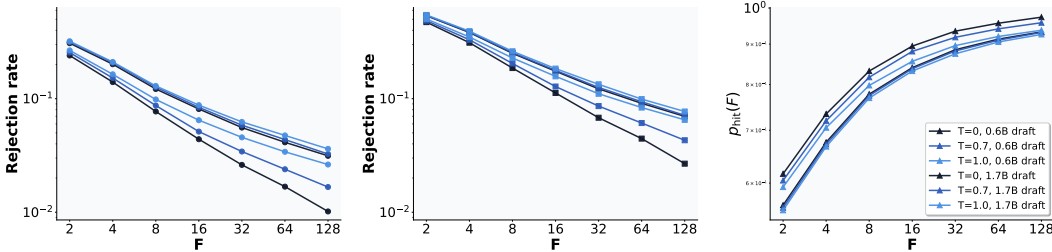

Figure 10: Rejection rate and cache hit rate scaling for Qwen-3 model family. Like the Llama-3 model family, we see it is an approximate power law in the fan-out, so that cache hit rates increase steadily as we grow $F$, the size of our cache.

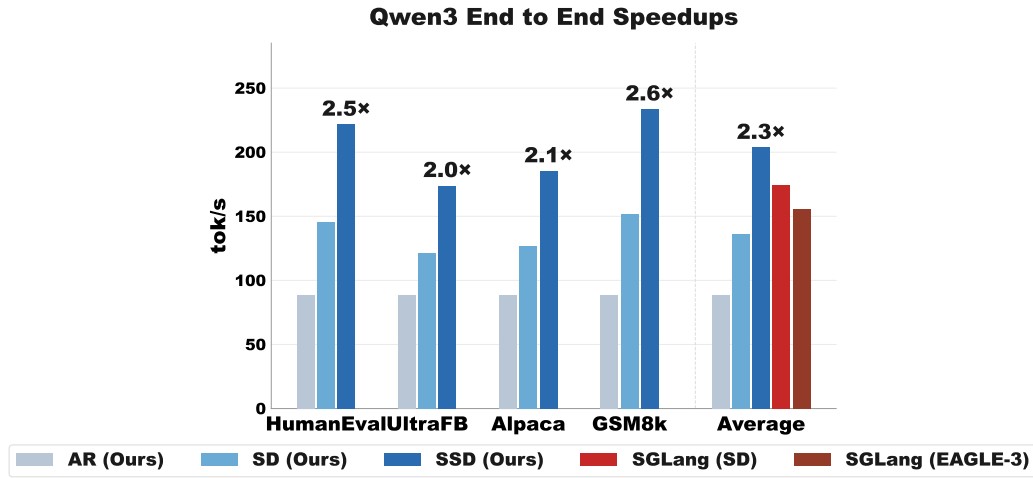

Figure 11: End-to-end decoding speed comparison of SSD compared to SD and standard autoregressive decoding on Qwen-3 32B across four datasets.

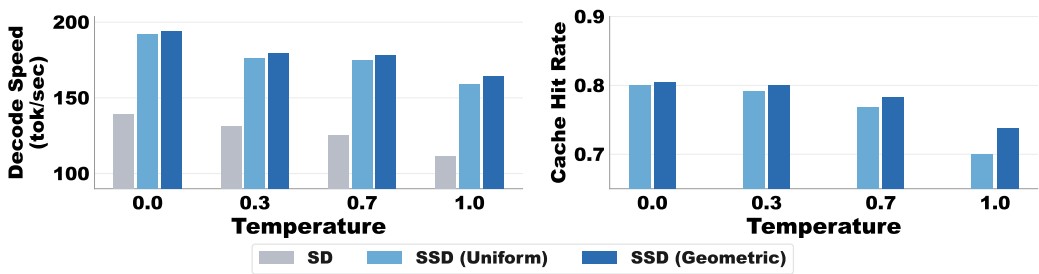

Figure 12: End-to-end speed for Qwen-3 model family compared to synchronous baselines.

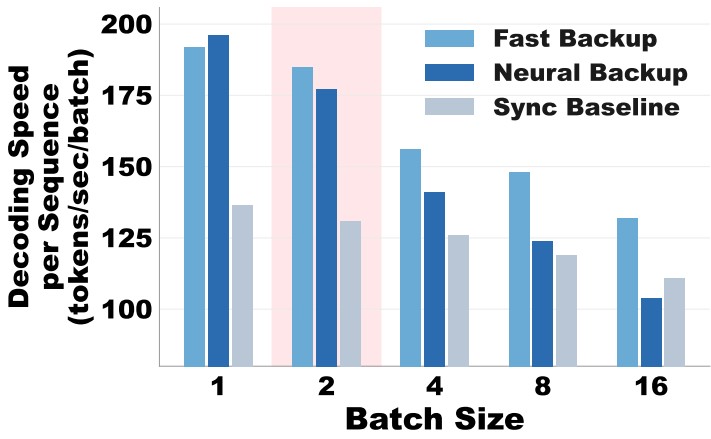

Figure 13: Batch size scaling of Qwen-3 models compared to synchronous baselines.

