# OpenReview forum: "Speculative Speculative Decoding"
_ICLR.cc/2026/Conference — ICLR 2026 Poster_

### Official Review · Reviewer_QxMH · 2025-10-23

**Soundness:** 4
**Presentation:** 4
**Contribution:** 4
**Rating:** 6
**Confidence:** 4

**Summary:**

This paper introduces ​Speculative Speculative Decoding (SSD)​, a novel framework that parallelizes the drafting and verification phases in speculative decoding. By pre-speculating likely verification outcomes during ongoing target model verification, SSD eliminates the sequential dependency between drafting and verification, achieving significant latency reductions.

**Strengths:**

1 The idea of speculating on speculationis both creative and impactful. By leveraging idle draft-model compute during verification, SSD addresses a fundamental bottleneck in existing speculative decoding methods. The theoretical framework (Algorithm 1) is elegant and generalizable.

2 The paper provides strong theoretical analysis.

3 Comprehensive Empirical Validation.

**Weaknesses:**

1 While SSD reduces latency, the parallel speculation across multiple branches increases draft-model FLOPs. The paper should quantify this overhead more explicitly.

2 Although comparisons to standard SD are thorough, benchmarking against recent parallel SD methods (e.g., SwiftSpec) or tree-based decoding (e.g., SpecInfer) would strengthen the claims. Does SSD outperform these methods in high-batch or high-temperature regimes?

**Questions:**

1 How does SAGUARO’s memory footprint scale with batch size and fan-out? Could cache eviction policies improve efficiency?

2 Could the pre-speculation idea be combined with approximate drafting (e.g., EAGLE-3) for further gains?

---

> ### Author Response · Authors · 2025-11-27
> **Official Response by Authors**
>
> Thanks for your the awesome review! We’re glad you felt the soundness, presentation and contribution of our work was excellent. We’ll take your points line by line.
>
> ## __Weakness 1__
> > _While SSD reduces latency, the parallel speculation across multiple branches increases draft-model FLOPs. The paper should quantify this overhead more explicitly._
>
> __You are correct that SSD increases draft model FLOPS, and we include the exact calculation in the new Appendix C.__ In summary, the draft uses $O(KF)$ more FLOPs than regular SD drafting, where $K$ is the speculative lookahead, and $F$ the fan-out-factor for predicting recovery tokens. Importantly, however, LLM decoding is typically memory-bound, not compute-bound, which makes additional FLOPs essentially free in this regime.
>
> ## __Weakness 2__
> > _Although comparisons to standard SD are thorough, benchmarking against recent parallel SD methods (e.g., SwiftSpec) or tree-based decoding (e.g., SpecInfer) would strengthen the claims. Does SSD outperform these methods in high-batch or high-temperature regimes?_
>
> We are unable to compare directly to SwiftSpec as there is no code released and the technique relies on many proprietary kernels. __We now compare with EAGLE-3 in Figure 8, a SOTA token-tree method, and find SSD to be comfortably faster (by ~36% and ~44% on average for Llama-3.3-70B and Qwen3-32B).__
>
> Regarding how SSD compares with these baselines across batch sizes and temperatures: SSD’s speedups over existing parallel SD methods, as well as tree-based SD methods, increase as batch size and temperature vary for the reasons below.
>
> __Batch size.__ SwiftSpec uses a naive just-in-time fallback, which works at low batch sizes when the probability of at least one cache miss is low, but becomes suboptimal at higher batch sizes when this probability rises. __Corollary 16 and Theorem 17 in Section 4.3 of our paper make this suboptimality precise.__ In the high-batch regime, token-tree methods like SpecInfer and EAGLE3 quickly struggle because verification time balloons as the compute done by the large verifier grows. SSD gets around these issues by adapting fallback strategies with batch size, and only increasing draft FLOPS, not verifier FLOPS.
>
> __Temperature.__ SwiftSpec only considers greedy decoding. As temperature rises, cache misses and an optimal fallback strategy become important considerations for the reasons above. SpecInfer struggles at low temperature because it can resample rejected tokens many times, using large amounts of verifier compute on these tokens without gain. __Saguaro sampling (Section 4.2) balances the fact that cache hit rates falls quickly at high temperature but acceptance rates fall slowly. The ability to trade them off in this subtle way allows for Saguaro to do very well at both low and very high temperatures (see Figure 6).__
>
> ## __Question 1__
> > _How does SAGUARO’s memory footprint scale with batch size and fan-out? Could cache eviction policies improve efficiency?_
>
> The memory on the draft scales linearly with both batch-size and fan out, but the total size is very small overall (megabytes) compared to GPU HBM (gigabytes), and is already refreshed every round. __We have added an explicit calculation of the memory usage on the draft in the new Appendix C.__
>
> ## __Question 2__
> > _Could the pre-speculation idea be combined with approximate drafting (e.g., EAGLE-3) for further gains?_
>
> We can absolutely combine SSD with Eagle-3! In fact, we are in the middle of implementing this combination, and __have outlined the systems design for how to do this in the new Appendix E.__ By our calculation, we should be able to attain at least 33% speedup over SSD by combining with EAGLE-3. We shared this calculation with Reviewer h3mq, and copy it here for convenience:
> > We now provide a simple back-of-the-envelope calculation on the expected speedups from combining SSD and EAGLE-3, estimating that we could attain at least ~33% speedup in the case of Llama-70B. We note that EAGLE-3 often uses a single-layer draft model as opposed to our current Llama-3.2-1B draft with 16 layers. In our measurements, this can reduce the draft forward pass latency from ~2ms (current) to less than 1ms. This allows us to at least double (e.g., from 5 to 10 for Llama-70B) the number of draft forward passes we can do during the verification, and thus increase the expected number of accepted tokens from ~4.2 to ~5.6 for Llama-70B (assuming the acceptance rate of EAGLE-3 matches that of Llama-3.2-1B). Thus, even without leveraging the improved acceptance rates of EAGLE-3 relative to standalone draft models, we expect to theoretically attain at least a 33% speedup (over our already SOTA tokens/sec) of our method end-to-end when SSD is combined with EAGLE-3!
>
> Please let us know if we can answer any other questions, and we appreciate the review. If we have addressed the questions or concerns you had, we would be delighted if you consider raising your score.

---

### Official Review · Reviewer_Jb3k · 2025-10-31

**Soundness:** 3
**Presentation:** 1
**Contribution:** 3
**Rating:** 6
**Confidence:** 4

**Summary:**

This paper introduces Speculative Speculative Decoding (SSD) — a new framework that parallelizes both speculation and verification in a distributed manner. While typical SD alternates executions between drafter and verifier, SSD removes this dependence and performs drafting and verification concurrently and asynchronously. In SSD, while verification for round T is ongoing, the draft model predicts the likely verification outcomes and pre-speculates corresponding continuations.

**Strengths:**

- This paper has novelty in terms of serving the draft and target models asynchronously.
- This paper has clear derivations and proofs (Theorem 7, 12, 17), which make the framework analytically sound.
- The author has conducted extensive benchmarks on multiple model families, detailed ablations (fan-out, sampling, fallback), and clear visualizations.

**Weaknesses:**

- The flow of the paper can be improved, e.g. Section 4 shows the results first and then explains the algorithms
- Can give a better flowchart of the workflow in Figure 1, e.g. label the step index
- This method requires an independent GPU for the speculator, this can increase the cost of serving and system complexity (e.g. heterogeneous GPUs)

**Questions:**

- While the deployment of LLMs typically runs on $2^n$ GPUs, SSD actually requires $2^{n} + 1$ GPUs. How can users deploy the SSD for serving efficiently? e.g. if using homogeneous GPUs, using a Hopper GPU for only the small draft model can be an overkill as much of the compute and memory are not saturated. If using heterogeneous GPUs (i.e. running the speculators on another type of GPU), how much impact will the communication between speculators and verifiers be?

---

> ### Author Response · Authors · 2025-11-27
> **Official Response by Authors**
>
> Thank you for your careful review. We have updated our manuscript in response to your comments and highlighted new portions in red. We respond to your comments inline:
>
> ## __Weakness 1__
> > _The flow of the paper can be improved, e.g. Section 4 shows the results first and then explains the algorithms._
>
> Thanks for this feedback. __We have now moved the end-to-end results (including new comparisons to SGLang with EAGLE-3) to Section 5, after the method has been introduced, as suggested.__
>
> ## __Weakness 2__
> > _Can give a better flowchart of the workflow in Figure 1, e.g. label the step index._
>
> __We have updated Figure 1 to now include the step index, as requested.__ Hopefully this makes the figure clearer. We’d love to hear any additional suggestions for how to improve this figure.
> ## __Weakness 3__
> > _This method requires an independent GPU for the speculator, this can increase the cost of serving and system complexity (e.g. heterogeneous GPUs)_
>
> __Our method actually does much better in cost, where the relevant metric is tokens/s/GPU.__ To illustrate this, we measured the performance of SGLang w/ SD on both TP=4 and TP=8 with Llama-3.3-70B. TP=8 is actually slightly slower since the compute per GPU is already small and there is more communication overhead in the forward pass. Thus in the end the tok/s/GPU (cost) are:
> - Llama-3.1-70B (SD, TP=4) ~ 138/4 = 34.5
> - Llama-3.1-70B (SD, TP=8) ~ 131/8 = 16.3
> - SSD (TP=4 + 1 draft GPU): ~ 230/5 = 46
>
> As we can see, SSD is the most economical.
>
> ## __Question 1__
> > _While the deployment of LLMs typically runs on $2^n$ GPUs, SSD actually requires $2^n + 1$ GPUs. How can users deploy the SSD for serving efficiently? e.g. if using homogeneous GPUs, using a Hopper GPU for only the small draft model can be an overkill as much of the compute and memory are not saturated. If using heterogeneous GPUs (i.e. running the speculators on another type of GPU), how much impact will the communication between speculators and verifiers be?_
>
> __The GPU used for speculation is in fact saturated, even if we use an H100. This is because we leverage the full capacity of the GPU to pre-speculate in parallel for as many verification outcomes as possible, decoding many more tokens than are verified on the draft model. We have added a concrete calculation of the extra FLOPs the draft model must pay compared to the target in our setup in Appendix C to make this clearer. Thus, there is no specific pressure to use heterogenous compute implied by our method.__
>
> Regarding communication, we find the overhead to be negligible, given that the only communication between draft and target is the verification outcome to the speculator once per round (tokens and logits). __We include an exact breakdown of how much data is sent and the time requirements in the new Appendix C.__
>
> Thank you for the perceptive questions. We believe we have revamped the diagrams/section to address presentation issues and explicitly addressed your technical concern. Let us know if we can help with anything else. Otherwise, we would appreciate it if you consider raising your score.

---

### Official Review · Reviewer_h3mq · 2025-10-31

**Soundness:** 3
**Presentation:** 3
**Contribution:** 3
**Rating:** 6
**Confidence:** 4

**Summary:**

This paper introduces Speculative Speculative Decoding (SSD), a framework that parallelizes the drafting and verification stages of speculative decoding by having the draft model predict likely verification outcomes in advance and pre-speculate for them. The authors propose SAGUARO, an optimized SSD algorithm that addresses three key challenges: accurately predicting verification outcomes, balancing cache hit rate with token acceptance rate, and handling cache misses efficiently. SAGUARO achieves up to 2x speedup over standard speculative decoding and up to 5x over autoregressive decoding on open-source inference engines.

**Strengths:**

1. The paper presents a principled and theoretically grounded approach to overlapping drafting and verification, effectively eliminating drafting latency through pre-speculation.

2. SAGUARO introduces practical and novel techniques—geometric fan-out allocation, cache-aware sampling, and adaptive fallback strategies—that collectively address core challenges in SSD.

3. The evaluation is comprehensive, covering diverse models, datasets, temperatures, and batch sizes, and shows significant end-to-end speedups while maintaining compatibility with existing acceleration methods.

**Weaknesses:**

1. The authors do not compare their method with existing SOTA methods (e.g., EAGLE2/3). They argue that their method is orthogonal to EAGLE and can be combined with it. However, the paper does not report the performance of their method when combined with EAGLE3. I would like to know what performance improvement could be achieved if they were combined.

2. The authors do not compare their method with existing Token Tree SPD methods, which I believe is a missing baseline. The proposed method (SSD) has the draft model send only a single speculative sequence to the verifier, and while the verifier validates that sequence, the draft model computes other possible branches in parallel. In contrast, Token Tree SPD methods have the draft model generate an entire "tree" structure at once, providing multiple candidate tokens at each position, and the verifier must validate all branches and nodes of this tree in a single forward pass. I consider these to be two similar solution approaches, and therefore the authors should compare these two types of methods. For example, they could replace the Token Tree SPD component in EAGLE3 with their SSD method and report whether the performance increases or decreases.

**Questions:**

1. I would like the authors to provide the wall time for their method. For example, while the target model is performing Verify, how many tokens can the draft model generate in that same amount of time?

2. I would like to know the number of accepted tokens for your method. Compared to EAGLE3, does your method accept more or fewer tokens per round?

I will adjust my score based on the author's rebuttal.

---

> ### Comment · Reviewer_h3mq · 2025-11-25
>
> Dear Authors,
>
> Thank you again for your submission. I would like to follow up on the questions and concerns I raised in my previous review. So far, I have not received a response or clarification. To fairly assess the contribution of your paper, I would appreciate it if you could address these points.
>
> In addition, I would also appreciate it if you could respond to the public comments related to your paper. These clarifications will help ensure a more accurate and fair evaluation. Otherwise, the lack of response may affect my final assessment and score.

---

> ### Author Response · Authors · 2025-11-27
> **Official Response by Authors (Part 1)**
>
> Thank you for the detailed review and close-read of our paper! We respond to your weaknesses/questions inline below, and have made updates to the manuscript (highlighted in red in the new version) as a result of your feedback. Please also feel free to see our response to the public comment above about related work.
>
> ## __Weakness 1__
> > _The authors do not compare their method with existing SOTA methods (e.g., EAGLE2/3). They argue that their method is orthogonal to EAGLE and can be combined with it. However, the paper does not report the performance of their method when combined with EAGLE3. I would like to know what performance improvement could be achieved if they were combined._
>
> __We have added an explicit comparison with the SOTA EAGLE3 implementation in SGLang in Figure 8, finding that we significantly outperform it (by ~36% and ~44% on average for Llama-3.3-70B and Qwen3-32B, respectively).__ To substantiate our claim that our method can also be combined with EAGLE-3, we are in the process of reimplementing EAGLE-3 in our optimized inference engine in a way that works with SSD. This is nontrivial from an engineering perspective because it requires passing target activations to the draft over NCCL and doing careful accounting of KV cache context when asynchronously drafting to track whether we are conditioning on draft vs target activations. But it is in progress, thanks for bringing this up! __We have added a detailed systems design for exactly how EAGLE-3 and SSD are combined in Appendix E.__
>
> We now provide a simple back-of-the-envelope calculation on the expected speedups from combining SSD and EAGLE-3, estimating that __we could attain at least ~33% speedup over SSD in the case of Llama-70B.__ We note that EAGLE-3 often uses a single-layer draft model as opposed to our current Llama-3.2-1B draft with 16 layers. In our measurements timing the forward passes of both draft models, this can reduce the draft forward pass latency from ~2ms (current) to less than 1ms. This allows us to at least double (e.g., from 5 to 10 for Llama-70B) the number of draft forward passes we can do during the verification, and thus increase the expected number of accepted tokens from ~4.2 to ~5.6 for Llama-70B (assuming the acceptance rate of EAGLE-3 matches that of Llama-3.2-1B). Thus, even without leveraging the improved acceptance rates of EAGLE-3 relative to standalone draft models, we expect to theoretically attain at least a 33% speedup (over our already SOTA tokens/sec) over SSD end-to-end when combining it with EAGLE-3!
>
> ## __Weakness 2__
> > _The authors do not compare their method with existing Token Tree SPD methods, which I believe is a missing baseline. The proposed method (SSD) has the draft model send only a single speculative sequence to the verifier, and while the verifier validates that sequence, the draft model computes other possible branches in parallel. In contrast, Token Tree SPD methods have the draft model generate an entire "tree" structure at once, providing multiple candidate tokens at each position, and the verifier must validate all branches and nodes of this tree in a single forward pass. I consider these to be two similar solution approaches, and therefore the authors should compare these two types of methods. For example, they could replace the Token Tree SPD component in EAGLE3 with their SSD method and report whether the performance increases or decreases._
>
> Thank you for pointing out the missing comparisons to token-tree verification SD methods, and the relationship between SSD and these methods.
>
> As mentioned above, __we have now added an explicit comparison between SSD and SGLang with EAGLE-3 (a SOTA token-tree SD method + SOTA draft architecture), showing that SSD meaningfully outperforms this strong baseline (by ~36% and ~44% on average for Llama-3.3-70B and Qwen3-32B, respectively).__ Additionally, we are in the process of implementing exactly the ablation you propose where we replace the token tree component of EAGLE3 with SSD to measure the performance gains we estimated above.
>
> The key point to have in mind is that you can think of SSD as drafting a tree, but only verifying a linear sub-sequence within that tree during each target model forward pass. This means it sidesteps the need for expensive tree verification by the large and slow target model. __Our main point is that the overall framework is a general synchronous to asynchronous algorithm rather than one that specifically competes with or replaces the myriad of existing variations on (synchronous) speculative decoding that already exist.__

---

> ### Author Response · Authors · 2025-11-27
> **Official Response by Authors (Part 2)**
>
> ## __Question 1__
> > _I would like the authors to provide the wall time for their method. For example, while the target model is performing Verify, how many tokens can the draft model generate in that same amount of time?_
>
> __We added a table with wall-clock metrics (like inter-token latency and tokens/sec) in Appendix B.3.__ For example, we can attain between 200 and 257 tokens/sec for Llama-70B with SSD depending on the dataset. Here is a wall clock breakdown of what happens in one asynchronous speculation round under the hood in our current implementation so you have intuition.
> - Verification latency + overhead: ~18ms (16 ms verification forward pass, 2 ms overhead)
> - Draft does 5 forward passes in this time (hidden entirely on critical path while verification taking place):
> - Expected number accepted tokens with 85% acceptance (measured, under greedy decoding): ~4.15
> - Overall: 18ms for ~4.15 tokens → ~230 tok/s, which is what we see empirically.
>
> ## __Question 2__
> > _I would like to know the number of accepted tokens for your method. Compared to EAGLE3, does your method accept more or fewer tokens per round?_
>
> For Llama-70B, we attain around 4 accepted tokens per round, which sits between EAGLE-2 (3.5 reported) and EAGLE-3 (~5 reported). These differences come from the fact that EAGLE-3 can attain higher acceptance rates than Llama-1B, while also doing token tree drafting/verification to increase the expected number of accepted tokens. SSD has the same acceptance rate as ordinary SD but hides the draft latency entirely, so is still faster than EAGLE3 end to end. Two things to note here:
> - Our draft model actually has longer to draft (~18ms above) than drafting usually takes in SD. This means in principle we can draft for more steps so that our expected tokens per round is higher than EAGLE3. However, in practice, the use of a custom attention mask leads to slower forward passes on the draft so that our implementation only fits in 5 draft forward passes per speculation round (comparable to the number of forward passes in SD). This is an engineering detail, not a limitation of the algorithm itself, and suggests there is room for further systems optimizations to push our inference speeds (already SOTA) even further.
> - As mentioned earlier, SSD and EAGLE3 can be combined in a way we articulate in the new Appendix E, and are in the process of implementing.
>
> Thank you for your thoughtful review. We hope this substantive response helps you better understand the nuances of the work and would be happy to engage further, and otherwise would appreciate if you consider raising your score. Thank you.

---

### Official Review · Reviewer_kXox · 2025-11-04

**Soundness:** 2
**Presentation:** 1
**Contribution:** 2
**Rating:** 2
**Confidence:** 3

**Summary:**

This paper proposes to parallelize the draft and verifier models in speculative decoding by running them on separate hardware devices. The proposed approach demonstrates superior performance compared to traditional speculative decoding in the reported experiments.

**Strengths:**

1.	The proposed method outperforms standard speculative decoding (SD) in the reported experiments.
2.	Introducing parallelism between the draft and verifier models represents an interesting and promising direction for advancing speculative decoding.

**Weaknesses:**

1.	The algorithmic description is difficult to follow in the current paper organization. It is unclear how the proposed method preserves the original distribution, and while the paper provides extensive theoretical analysis, it offers limited explanation and intuition about the algorithm itself, particularly in Section 3.
2.	The paper lacks baseline comparisons against other speculative decoding (SD) approaches, making it hard to assess the relative performance and contribution of the proposed method.
3.	The paper does not include a dedicated experimental section—experiments appear only as a subsection—and there is insufficient detail regarding the setup, such as which models were evaluated and under what conditions.

**Questions:**

Could you improve the paper’s structure to make the algorithm easier for readers to follow?
Additionally, could you add more experiments to address the weaknesses noted in points 2 and 3, particularly by including comparisons with other speculative decoding methods and providing more detailed experimental setups?
Finally, could you clarify how your method preserves the original distribution, both conceptually and mathematically?

---

> ### Author Response · Authors · 2025-11-27
> **Official Response from Authors**
>
> We thank the reviewer for their time. We have rewritten parts of the manuscript to explicitly address your feedback, and addressed some misconceptions below. Changes are highlighted in red in the new manuscript.
>
> ## __Weakness 1__
> > _The algorithmic description is difficult to follow in the current paper organization. It is unclear how the proposed method preserves the original distribution, and while the paper provides extensive theoretical analysis, it offers limited explanation and intuition about the algorithm itself, particularly in Section 3._
>
> Thank you for sharing that you found the presentation of the SSD algorithm hard to understand, and that it was unclear to you why the algorithm is lossless (i.e., output distribution matches autoregressive decoding). __We have added intuition and further explanations to Section 3, which we also share here.__
>
> To provide intuition about the algorithm, and better understand why it is lossless, it is useful to think about why the classical technique of speculative execution can speed up CPU performance without changing the behavior of the code. Speculative execution uses idle compute resources to preemptively execute certain conditional blocks or pre-fetch data before it is known if they'll be needed, such that if they are eventually needed, the work will already be done. SSD is doing something analogous with LLM forward passes relative to ordinary SD. SSD uses available compute resources to pre-speculate for many possible verification outcomes, such that if any of those outcomes does occur, the next speculated token sequence will be ready immediately. This token sequence is then verified in the same exact way SD performs verification, thereby maintaining the lossless properties of standard SD. In the case where SSD did not prepare for the verification outcome that actually occurred, a synchronous ``fallback speculator'' is used to generate the speculation, which also maintains the lossless property of SD.
>
> We hope that this makes the algorithm much clearer, and we would be happy to answer any follow up questions. Thanks for bringing this up.
>
> ## __Weakness 2__
> > _The paper lacks baseline comparisons against other speculative decoding (SD) approaches, making it hard to assess the relative performance and contribution of the proposed method._
>
> Our baselines are very strong: our implementation of ordinary speculative decoding as a baseline is faster than that of SGLang with SD, which was the fastest of all the popular inference engines (vLLM, SGLang, TRT-LLM) we tried. __To compare with an even stronger baseline, __we have added a new comparison to SGLang with EAGLE-3 in Figure 8, and demonstrate that SSD outperforms this (previously) SOTA baseline by ~36% and ~44% on average for Llama-3.3-70B and Qwen3-32B, respectively.__ Also, it is possible to combine our method with EAGLE-3, and we are in the process of implementing this. We have added explicit discussion around how exactly the two methods can be implemented together in detail in Appendix E.__
>
> ## __Weakness 3__
> > _The paper does not include a dedicated experimental section—experiments appear only as a subsection—and there is insufficient detail regarding the setup, such as which models were evaluated and under what conditions._
>
> Thank you for this feedback. __We have now moved our end-to-end results to a standalone section (Section 5) per your suggestion.__ We have chosen, for now, to keep the more fine-grained experimental results validating the three main Saguaro optimizations embedded in their corresponding subsections, to make it easier for readers to assess the claims for that section while these (often complicated) ideas are fresh in the reader’s mind.
>
> __To see details about our experimental setup, we’d like to refer you to our “Setup” section at the beginning of Section 4 (further details in Appendix B.2), which is written specifically to answer your questions.__ We have highlighted these in red in the new manuscript to emphasize this in case you want to take a look.
>
> We believe we have addressed each concern or issue posed by the reviewer, and we would be delighted to answer any other outstanding questions, or otherwise invite the reviewer to consider raising their score. Thank you.

---

### Public Comment · ~Yuhao_Shen3 · 2025-11-18
**Public Comment**

Dear Authors,

I hope this email finds you well. I have carefully read your paper. I noticed that the motivation for SSD discussed in your introduction (specifically the three challenges) and the approach to multi-path draft generation bear a strong resemblance to an arXiv article [1]. Furthermore, I observed that both works conduct a theoretical analysis of rollback mechanisms for Parallel SD. However, I noticed that this paper [1] is not currently cited in your submission, nor is there a discussion or comparison with it. I believe SSD is an excellent piece of work on Parallel SD, supported by solid theoretical analysis. And if you explicitly cite this work and provide a detailed discussion or comparison regarding the differences, it would significantly clarify your contributions and strengthen the novelty of your paper.

Reference:

[1] "Speculative Decoding via Hybrid Drafting and Rollback-Aware Branch Parallelism." arXiv preprint arXiv:2506.01979 (2025).

Best regards,

---

> ### Author Response · Authors · 2025-11-27
> **Official Response from Authors**
>
> Thank you for bringing this paper to our attention, and our apologies for not being aware of it and citing it in our submission—__we have now added this citation and discussion to our substantially expanded related work section__ (see the newly attached manuscript, all changes highlighted in red). Indeed, SpecBranch has similar motivation to SSD, and some similar components: we can think of SpecBranch as (approximately) a special case of the SSD framework where only a single branching point is allowed, where the fallback speculator is equal to the regular speculator, and where the hyperparameters (branching point, number of branches, speculation length) are dynamically chosen.
>
> SSD and Saguaro build upon the idea in SpecBranch of pre-speculating alternate token paths in the case of a rejection (in parallel to verification), though they take a somewhat different approach to doing so. The focus of most of our paper is in identifying and studying several design considerations for a parallel speculation system beyond the considerations in SpecBranch. These include:
> - Allowing for an arbitrary number of branch points (thereby greatly increasing the probability of one of the branches being “correct” and thus improving cache hit rate) in Section 4.1.
> - Identifying a new and extremely subtle tradeoff between cache hit rate and acceptance rate, and optimizes the draft sampling distribution in light of this tradeoff in Section 4.2. Exploiting this leads to large speedups at high temperature (Figure 6) when speculative decoding techniques usually fail.
> - Carefully choosing the “fallback” strategy for the case where none of the branches were correct (important for batch size > 1 and temperature > 0) in Section 4.3.
> - Providing theoretical results and bounds around what is the optimal strategy for each: predicting recovery tokens, recovering from cache misses, reasoning about the Pareto frontier in cache hit vs acceptance rate, and about the expected and maximal gains from each of these design choices both in isolation and end to end.
>
> In the end, our algorithm and implementation attains a new state-of-the-art average of __~230 tok/sec__ (Llama-70B/1B, TP=4, H100), compared to __~26 tok/sec__ for SpecBranch in a similar but not identical setup (Llama-70B/8B, TP=4, A100). To achieve this, we had to implement our design choices in a way that works efficiently with staple parts of the modern inference stack: PagedAttention, continuous batching, CUDAgraphs, and torch compilation, each of which make a naïve implementation of asynchrony untenably slow. __We have added a substantial discussion of these important systems design details in Appendix B.1.__ We will open source the engine we wrote at the time of conference decisions to avoid violating anonymity. We have definitely built on important prior work, and have significantly expanded our discussion of related work: thank you again for bringing this very interesting and relevant work to our attention, we are glad to cite and discuss it!

---

> ### Public Comment · ~Yuhao_Shen3 · 2025-11-28
>
> I appreciate your detailed response. The revisions have elevated the paper's novelty and contribution.
> However, I noticed that SpecBranch does not utilize Tensor Parallelism (TP) for acceleration. The implementation appears to be based solely on the transformers.
> To my knowledge, nano-PEARL [1] is the only open-source system framework for parallel speculative decoding and I am one of the contributors. Therefore, I have a question regarding SSD: Upon which framework is your system implementation built? Did you complete a full system-level implementation (page attention, cuda graph, torch compile)entirely during the rebuttal phase? I look forward to the open-sourse of SSD.
> [1]https://github.com/smart-lty/nano-PEARL

---

> ### Author Response · Authors · 2025-12-03
>
> We are glad to hear that you have appreciated our response and paper revisions, and how they better highlight the paper's novelty and contribution. We are glad to answer your follow-up questions here:
> - __Regarding tensor parallelism__: We were referring to how inference in SpecBranch for the 70B model was performed across 4 GPUs, which is the same number of GPUs we use for SSD. We based this on the details from Appendices E.2 and E.3 of the SpecBranch paper, which specify that it runs “_70B models on four devices_” using “_bfloat16 format for optimized GPU inference without quantization_” on “_NVIDIA A100-PCIE-40G GPUs_”. Our implementation uses tensor parallelism; HuggingFace defaults to pipeline parallelism, but we believe this to be a fair comparison because both serve the same model at the same precision on the same number of devices.
> - __Regarding the framework upon which SSD was built__: We implement our engine from scratch in pure PyTorch making use only of existing attention kernels like FlashAttention/FlashInfer. The entire engine (including support for PagedAttention, continuous batching, CUDAgraphs, torch compilation, etc) was written before submission, not during rebuttals. That is what was used for SSD tok/sec benchmarking. The systems design portion of the Appendix was also present in the originally submitted manuscript, we simply expanded it and highlighted it during the rebuttal to make it clearer.
>
> Thank you so much for your interest in our paper and for your excitement about us open-sourcing our engine at the time of conference decisions.

---

### Meta-Review · Area_Chair_wC7Y · 2025-12-20

**Summary:**

**Paper summary.** This paper proposes Speculative Speculative Decoding (SSD). The goal is to remove the remaining sequential dependency in speculative decoding: in standard SD, drafting and verification alternate, so drafting often sits idle while the target model verifies. SSD overlaps these two phases by having the draft model predict likely verification outcomes and pre-compute the next speculations asynchronously. The paper presents a concrete system (Saguaro), proofs for correctness, and end-to-end throughput gains on multiple model families and settings.

**What happened in the discussion.** Reviewers agreed the idea is interesting and the evaluation is broad, but raised three concrete issues: (1) the paper must compare to strong recent baselines (EAGLE-2/3, token-tree methods, other parallel SD systems), (2) the method increases draft-side compute and may stress memory/KV-cache, so overhead should be quantified with wall-time metrics, and (3) SSD may require an extra GPU/device for the speculator, which affects deployment cost. In the forum, the authors added comparisons (including an explicit EAGLE-3 comparison in an updated benchmark), added wall-clock metrics and a memory/overhead calculation, improved the paper organization/figure, and clarified how SSD can be combined with approximate drafting (they also outline an SSD+EAGLE-3 design and estimate expected gains).

**My assessment as AC.** The main contribution is a clean systems idea (asynchronous pre-speculation) backed by a detailed implementation and a solid correctness argument. The “extra device” requirement is a real trade-off, but the paper is transparent about it and provides enough data to judge the latency/throughput benefits. After the discussion, the biggest remaining weakness is that baseline coverage will never be fully complete in this fast-moving area; however, the added comparisons and wall-time reporting make the story credible. I also think the revised structure/intuition addresses the “hard to follow” complaint.

**Decision.** Accept (poster). The method is technically sound, the empirical evidence is strong, and the authors responded directly to the key reviewer concerns with concrete additional results and clarifications.

**Reviewer Concerns:**

- **Reviewer QxMH (rating 6, confidence 4)**: Asked for clearer quantification of extra compute/memory overhead and for stronger baseline comparisons; authors added an explicit overhead calculation and more baseline comparisons and discussion. **Status:** largely resolved.
- **Reviewer Jb3k (rating 6, confidence 4)**: Raised paper flow/figure clarity and the “extra GPU/device” deployment cost (2^n + 1) question; authors reorganized the paper and clarified deployment and communication considerations. **Status:** partially resolved (cost trade-off remains, but is now clearly explained).
- **Reviewer h3mq (rating 6, confidence 4)**: Asked for direct comparisons to EAGLE2/3 and token-tree SD, plus wall time and accepted-token metrics; authors added an EAGLE-3 comparison, added wall-clock measurements, and described SSD+EAGLE-3 combination plans. **Status:** mostly resolved.
- **Reviewer kXox (rating 2, confidence 3)**: Found the algorithm hard to follow, unclear why it is lossless, and requested more baseline comparisons; authors added intuition for losslessness and improved organization/figures and comparisons. **Status:** partially resolved; readability is much improved but this reviewer may still dislike the systems trade-offs.

**Reviewer Scores:**

- **QxMH (6,4)**: Likely unchanged (strong accept already).
- **Jb3k (6,4)**: Likely unchanged.
- **h3mq (6,4)**: Likely unchanged or slightly higher given added EAGLE-3 comparison and wall-clock table.
- **kXox (2,3)**: I expect the score could move toward borderline (~4) after the added explanations/comparisons, but the reviewer did not explicitly confirm a score change.

---

### Decision · Program_Chairs · 2026-01-26

Accept (Poster)